# Glia-neuron interactions underlie state transitions to generalized seizures

Carmen Diaz Verdugo [1,2,3,10], Sverre Myren-Svelstad [1,4,5,10], Ecem Aydin[1,6], Evelien Van Hoeymissen[1,3], Celine Deneubourg [1,3], Silke Vanderhaeghe[1,3], Julie Vancraeynest[1,3], Robbrecht Pelgrims[1], Mehmet Ilyas Cosacak[7], Akira Muto [8], Caghan Kizil [7], Koichi Kawakami[8], Nathalie Jurisch-Yaksi [1,5,9,11] & Emre Yaksi [1,2,3,5,11]

Brain activity and connectivity alter drastically during epileptic seizures. The brain networks shift from a balanced resting state to a hyperactive and hypersynchronous state. It is, however, less clear which mechanisms underlie the state transitions. By studying neural and glial activity in zebrafish models of epileptic seizures, we observe striking differences between these networks. During the preictal period, neurons display a small increase in synchronous activity only locally, while the gap-junction-coupled glial network was highly active and strongly synchronized across large distances. The transition from a preictal state to a generalized seizure leads to an abrupt increase in neural activity and connectivity, which is accompanied by a strong alteration in glia-neuron interactions and a massive increase in extracellular glutamate. Optogenetic activation of glia excites nearby neurons through the action of glutamate and gap junctions, emphasizing a potential role for glia-glia and glia-neuron connections in the generation of epileptic seizures.

[1] Faculty of Medicine and Health Sciences, Kavli Institute for Systems Neuroscience and Centre for Neural Computation, Norwegian University of Science and Technology, 7030 Trondheim, Norway. [2] Neuro-Electronics Research Flanders, 3001 Leuven, Belgium. [3] KU Leuven, 3000 Leuven, Belgium. [4] Faculty of Medicine and Health Sciences, Department of Neuromedicine and Movement Science, Norwegian University of Science and Technology, 7030 Trondheim, Norway. [5] Department of Neurology and Clinical Neurophysiology, St. Olav's University Hospital, 7030 Trondheim, Norway. [6] Department of Biomedical Engineering, İzmir Katip Çelebi University, 35620 İzmir, Turkey. [7] German Center for Neurodegenerative Diseases (DZNE) Dresden, Helmholtz Association / Technische Universität Dresden, Center for Molecular and Cellular Bioengineering (CMCB), Center for Regenerative Therapies Dresden (CRTD), Dresden 01307, Germany. [8] Laboratory of Molecular and Developmental Biology, National Institute of Genetics, and Department of Genetics, SOKENDAI (The Graduate University for Advanced Studies), Mishima, Shizuoka 411-8540, Japan. [9] Faculty of Medicine and Health Sciences, Department of Clinical and Molecular Medicine, Norwegian University of Science and Technology, 7030 Trondheim, Norway. [10] These authors contributed equally: Carmen Diaz Verdugo, Sverre Myren-Svelstad. [11] These authors jointly supervised this work: Nathalie Jurisch-Yaksi, Emre Yaksi. Correspondence and requests for materials should be addressed to N.J.-Y. (email: nathalie.jurisch-yaksi@ntnu.no) or to E.Y. (email: emre.yaksi@ntnu.no)

Epilepsy is a group of neurological disorders characterized by recurrent seizures[1]. When the brain switches from a resting state into a seizure, dramatic transitions occur leading to abnormally synchronous brain activity. Profound changes in neural activity and connectivity were observed during seizure initiation and propagation[2]. High-frequency oscillations and breakdown of inhibition are often considered as the key triggers of ictal discharges[3–8]. Once the inhibition surrounding the epileptic focus breaks down, excessive neural activity spreads to highly connected hubs and propagates to the rest of the brain[4]. These hubs display increased synchrony and connectivity during seizures[9] and play an important role in seizure propagation by acting as gatekeepers between seizure foci and the global brain network[10]. Focal low amplitude and high-frequency neural activity evolves into large amplitude and low-frequency activity, as localized seizures propagate[11–13]. Taken together, the initiation and propagation of seizures are often viewed as transitions between brain states. For example, epilepsy patients show a transition from a balanced baseline connectivity state into a temporary hypersynchronous connectivity state[13]. It is, however, unclear how such profound changes in neural activity and connectivity can occur rapidly during state transitions of generalized seizures[13,14].

The important role of glia in modulating synaptic connectivity and neural excitability is now well accepted[15,16]. Astrocytes, the largest group of glial cells, regulate the availability of neurotransmitters and ions at the synaptic cleft, especially glutamate and potassium[16–18]. Moreover, astrocytes are highly connected through gap junctions, and form a functional syncytium, which effectively redistributes ions and neurotransmitters across large distances in the brain[19,20]. Interestingly, transgenic mice with a deficiency in the astrocytic gap junction coupling show spontaneous seizures[21]. Alterations of astrocytic coupling and clearance of potassium are linked with temporal lobe epilepsy, both in rodents and human patients[21–23]. Precisely how glia-neuron interactions are involved in initiation and propagation of epileptic seizures remains an open question with potential therapeutic applications.

To address this question, we characterized the state transitions of neural and glial activity and connectivity in pharmacological-induced seizure models of zebrafish larvae[24–26]. The brains of transparent zebrafish larvae combined with two-photon calcium imaging allowed us to monitor the activity and functional connectivity of thousands of individual neurons and glia, in vivo, with unprecedented spatiotemporal resolution, across several brain regions[27]. We show that during the preictal state, the neural synchrony increases only locally, while the gap junction coupled glial networks are activated and synchronized across large distances, independent of neural activity and synchrony. We also observe that glial activity bursts during preictal state correspond to dampening of neural activity. At the initiation of a generalized seizure, when a strong surge of both glial and neural activity occur, we observe a rapid transition in the functional connectivity of neural and glial networks, as well as a sudden increase in extracellular glutamate. Finally, we show that activation of glial networks leads to a strong increase in neural activity through the action of glutamate and gap junctions. Taken together, these interactions could underlie the rapid transition from a preictal state to a generalized seizure, which goes beyond neural connectivity rules. We propose that alterations in glia-neuron interactions underlie state transitions that lead to generalized epileptic seizures.

## Results

**Calcium imaging reliably reports epileptic neural activity**. To monitor the seizures across multiple brain regions with high temporal and spatial resolution, we first examined to what extent epileptic activity can be captured with genetically encoded calcium indicators in larval zebrafish. To this end, we measured the electrical activity and calcium signals in *Tg(elavl3:GCaMP6s)*[28] zebrafish larvae expressing *GCaMP6s* in all neurons simultaneously[29,30]. Seizures were induced through application of pentylenetetrazole (PTZ), a widely used proconvulsant agent that is a GABA$_A$ receptor antagonist[24,25,31] (Supplementary Fig. 1). Previous reports have shown that electrical activity can be reliably measured by local field potential (LFP) recordings at high frequency with a micro-electrode inserted in the midbrain[24,25]. Hence, we performed LFP recordings in immobilized *Tg(elavl3: GCaMP6s)* larvae under an epifluorescence microscope (Fig. 1a). The neural activity reported by the calcium indicator clearly reflected the electrical activity measured by LFP (Fig. 1b). Importantly, calcium imaging enabled the detection of neural events both prior to drug exposure (baseline), prior to the generalized seizure (preictal) and during the generalized seizure, defined as the major epileptic activity involving all brain regions simultaneously (Fig. 1c). To provide further evidence that calcium imaging can capture changes in the epileptic network activity, we analyzed the oscillatory power of neural activity in electrical and calcium signals during these three periods. Since the onset of a generalized seizure varied across larvae, we evaluated the preictal period as the minute preceding the generalized seizure. Power spectral density analysis of LFP and calcium signals showed that the oscillatory changes of network activity can be detected by both techniques, and that there was an increase of oscillatory activity during the preictal period at frequencies less than 10 Hz (Fig. 1d–f). During generalized seizures we observed a reduction of power in high-frequency LFP activity (Fig. 1d, e), and increased power of low-frequency LFP and calcium activity (Fig. 1d–f). Mean square coherence between LFP and calcium signals indicated that calcium signals can effectively carry information from the preictal and ictal events mainly at low frequencies, but less effectively at high frequencies (Fig. 1g). Altogether, our data shows that calcium imaging allows us to detect both the generalized seizures that spread across the entire brain, but also the alterations in neural activity during preictal discharges.

**Brain regions are differentially recruited during seizures**. To investigate the neural circuits activity during transition of the brain from a healthy baseline state to an epileptic state, we performed two-photon calcium imaging of *Tg(elavl3:GCaMP6s)* zebrafish larvae across multiple brain regions with single cell resolution. We collected data from five anatomically defined brain areas: telencephalon, thalamus, optic tectum, cerebellum and brainstem (Fig. 2a, b, Supplementary Movie 1). Upon semiautomated cell detection, we obtained the neural activity of more than two thousand neurons per fish (Fig. 2a, b). To better understand the neural network transitions, we first quantified the number of active neurons and their average activity during seizure generation (ictogenesis). We compared the activity of three different time periods: 3 min prior to PTZ perfusion (baseline), 3 min prior to onset of the first seizure (preictal period), and 3 min after the seizure onset. We observed a significant increase in the overall ratio of active neurons during the transition from baseline to preictal period. During the generalized seizure, more than 90% of all measured neurons were active (Fig. 2c). Upon analysis of individual brain regions during the preictal period, we saw a significant increase in the ratio of active neurons only in the optic tectum and cerebellum, but not in the telencephalon, thalamus nor brainstem (Fig. 2d). Yet, during generalized seizures, all brain regions recruited significantly more active neurons (Fig. 2d).

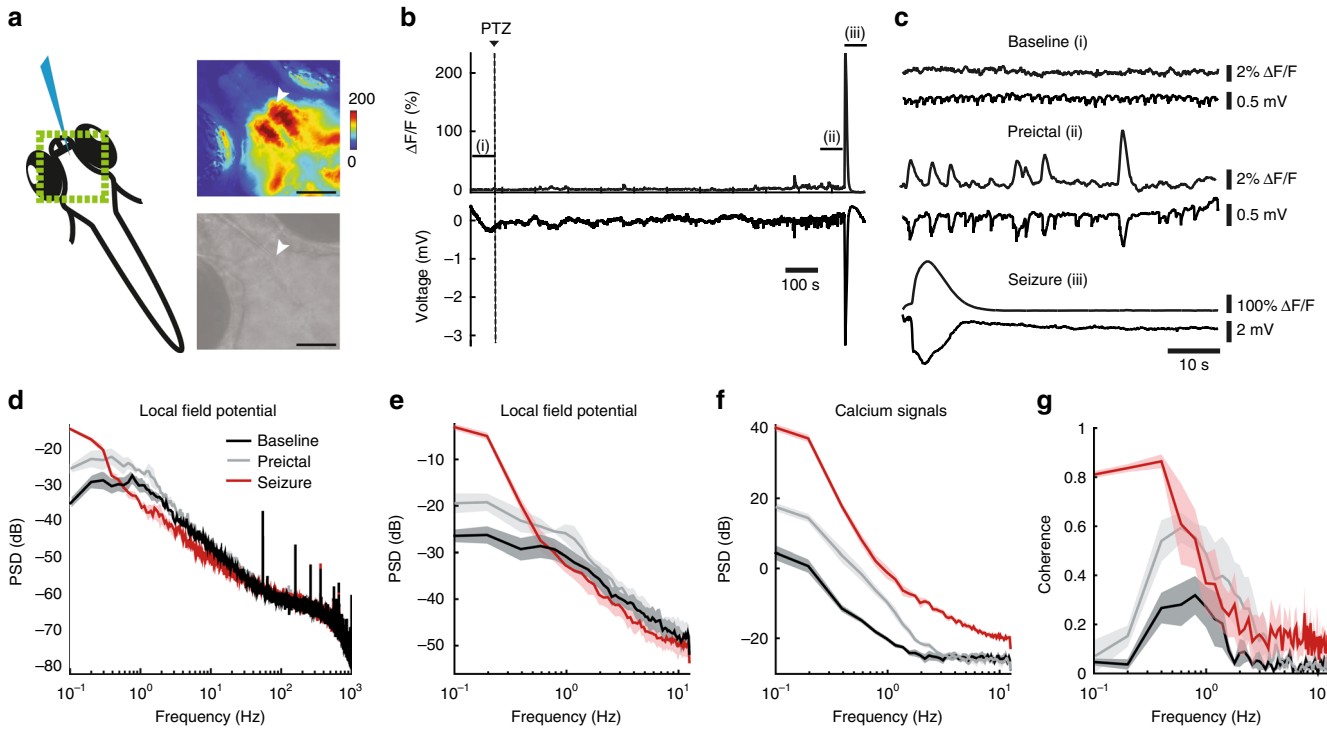

**Fig. 1** Calcium imaging reliably reports temporal components of neural activity during preictal and generalized seizure period. **a** Scheme representing simultaneous local field potential (LFP) recording and epifluorescence calcium imaging. Green box indicates the region recorded with calcium imaging. The top right image shows the fluorescence intensity change (ΔF/F) during a seizure, warmer colors indicate stronger activity. The bottom gray scale panel is a raw image of a zebrafish brain (7 days post-fertilization (dpf), *Tg(elavl3:GCaMP6s)*). White arrowhead indicates the electrode position in the optic tectum. Black bar reflects 100 μm. **b** Calcium signals (ΔF/F) recorded near the electrode (upper) and electrical activity recorded with the electrode (lower). Dashed line indicates the start of 20 mM pentylenetetrazole (PTZ) perfusion. **c** Enlargement of the three time windows marked in Fig. 1b. Baseline represents spontaneous activity (prior to drug exposure), preictal period is prior to the onset of the seizure, and the seizure is the first minute following the onset of the generalized seizure. **d** Power Spectral Density (PSD) of the LFP during the three time windows: baseline (black), preictal (gray) and seizure (red). **e** PSD of the LFP, decimated to 25 Hz recording rate. **f** PSD of the ΔF/F calcium signals recorded at 25 Hz. **g** Magnitude squared coherence between LFP and ΔF/F calcium traces (both at 25 Hz). Shaded regions denote s.e.m., n = 5 fish

Interestingly, the telencephalon was recruited significantly slower than most brain regions at the onset of the generalized seizures (Supplementary Fig. 2). Next, we quantified the activity of active neurons by calculating the area under the curve of the calcium events for individual neurons. While we detected an activity increase of more than tenfold in all brain regions during generalized seizures (Fig. 2e, f), throughout the preictal period we observed a significant increase in the activity of active neurons only in the optic tectum (Fig. 2f). Taken together, our results show that individual brain regions are differentially recruited during the preictal period, with significantly larger changes in the optic tectum and the smallest changes in the telencephalon. This is in line with differences in the distribution of GABAergic (*Tg(gad1:GFP)*)[32] versus glutamatergic neurons (*Tg(vglut2a: dsRED)*)[33] across the brain (Fig. 2g).

**The state transition to generalized seizure is abrupt**. The current knowledge suggests that a switch from a balanced connectivity state into a hypersynchronous connectivity state underlie generation of epileptic seizures. To test this hypothesis, we investigated the alterations in functional connectivity within and across brain regions during preictal and ictal periods. We calculated the pairwise Pearson's correlations to quantify the functional connectivity between neurons. To remove the influence of slow shifts in baseline on the correlations of neural activity we performed a running baseline subtraction. Neurons with highly synchronized calcium traces led to high correlations

(Supplementary Fig. 3A-3D). We observed only a small increase in correlations of neural activity between baseline and preictal period while comparing all neurons across the brain (Fig. 3a, b). During generalized seizures, as the entire brain became active, neurons were highly correlated to each other within and across brain regions (Fig. 3a–c). When brain regions were analyzed individually, there was a significant increase in correlations between neurons located within the optic tectum, cerebellum, and brainstem (Fig. 3b). Across brain regions we showed increase in correlations between thalamus and cerebellum, as well as between optic tectum and brainstem (Fig. 3c). During a generalized seizure all neurons and all brain regions were synchronized (Fig. 3b, c). Shuffling individual neural time series led to a disruption of these correlation patterns during all time windows (Supplementary Fig. 3E–3H). These findings are in line with the hypothesis that nearby neurons would be more likely to have functional connections, thereby showing stronger synchrony during baseline and preictal period. To test the relationship of distances between neurons and their functional connections, we visualized the correlations of neural activity with neural distance. During the baseline period, nearby neurons showed stronger correlations (Fig. 3d, black solid lines) when compared to spatially shuffled distributions, where neural positions were randomly assigned (Fig. 3d, black dotted lines). These results confirm stronger functional connectivity between nearby neurons and weaker connectivity between neurons that are more than few hundred micrometers away. During the preictal period, the synchrony between nearby neurons increased, in accordance with increased

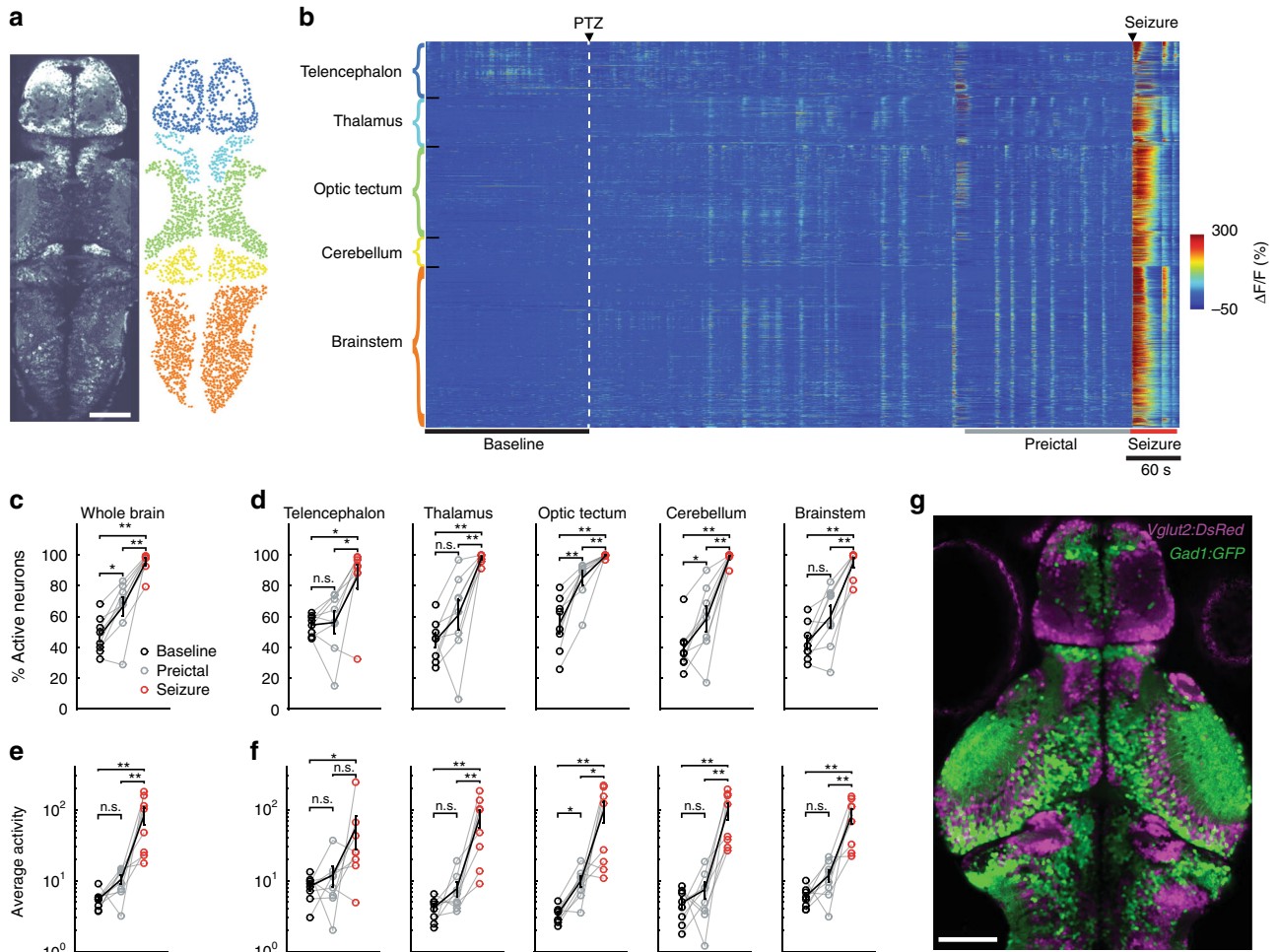

**Fig. 2** Brain regions are differentially recruited during epileptic activity. **a** An optical section of a zebrafish larva expressing *GCaMP6s* in all neurons, obtained by two-photon microscopy, dorsal view (left). Individual neurons (right) in color-coded brain regions: telencephalon (dark blue), thalamus (light blue), optic tectum (green), cerebellum (yellow), and brainstem (orange). **b** Activity of individual neurons (ΔF/F) over time, organized by brain region. White dashed line indicates the start of 20 mM pentylenetetrazole (PTZ) perfusion. Warmer colors indicate stronger activity. **c, d** Percentage of active neurons (>3std_baseline) in the whole brain (**c**), and per brain area (**d**) during baseline (black), preictal (gray) and seizure periods (red). **e, f** Average activity of the active neurons, defined by the area under the curve of the ΔF/F trace, in the whole brain (**e**) and per brain area (**f**). **g** Confocal image of *Tg(gad1:GFP);Tg(vglut2a:dsRED)* double transgenic zebrafish larva showing glutamatergic (magenta) and GABAergic (green) neurons. White bars reflect 100 μm. **p = < 0.01, *p = < 0.05, ns = not significant, Wilcoxon signed-rank test. Error bars (**c-f**) represent the s.e.m. of n = 8 fish, where 2012.6 ± 238.7 (mean ± SEM) neurons/fish were analyzed

connectivity and synchrony within several individual brain regions shown in Fig. 3b. On the contrary, the synchrony between neurons that are more than only few hundred micrometers apart showed little or no change of connectivity, and connectivity levels were similar to spatially shuffled neurons (Fig. 3d, gray lines). During the generalized seizures, however, we observed a dramatic increase in neural synchrony even between those neurons that are more than several hundred micrometers apart (Fig. 3d, red lines). These findings suggest a drastic alteration of neural connectivity and synchrony during generalized seizures that is largely independent of spatial distances between neurons, which might be mediated by processes beyond typical neuron-to-neuron communication.

To further investigate this drastic alteration of neuron-to-neuron communication, we studied the nature of functional connectivity between all neurons across the brain by visualizing positive and negative correlations in the form of a histogram. In line with our findings based on average correlations across all neurons (Fig. 3b), we observed only a slight increase in positive pairwise correlations between neurons from the baseline to the

preictal period (Fig. 3e). To our surprise, during generalized seizures most negative correlations between neurons were eliminated, indicating a major rearrangement of neural connectivity rules. We hypothesize that this rearrangement could in principle be either due to gradual elimination of inhibitory connections through the antagonistic action of PTZ on GABA_A receptors or due to processes that inject massive excitation into the neural network and synchronize those neurons with inhibitory connections by simultaneously activating them. To investigate the time scale for this drastic alteration of neural connectivity, we measured the neural synchrony at different time points (seven intervals of 60, 10, or 3 s) before the seizure. We found that the rearrangement of neural synchrony was not a gradual process that is progressively morphing the neural connectivity rules, but instead a rapid process that reshapes the functional connectivity histograms, within a few seconds (Fig. 3f). These results suggest that it is unlikely that the alteration of neural connectivity is due to gradual elimination of inhibitory connections. Instead, our data is consistent with a rapid process that can deliver excitation to the entire nervous system

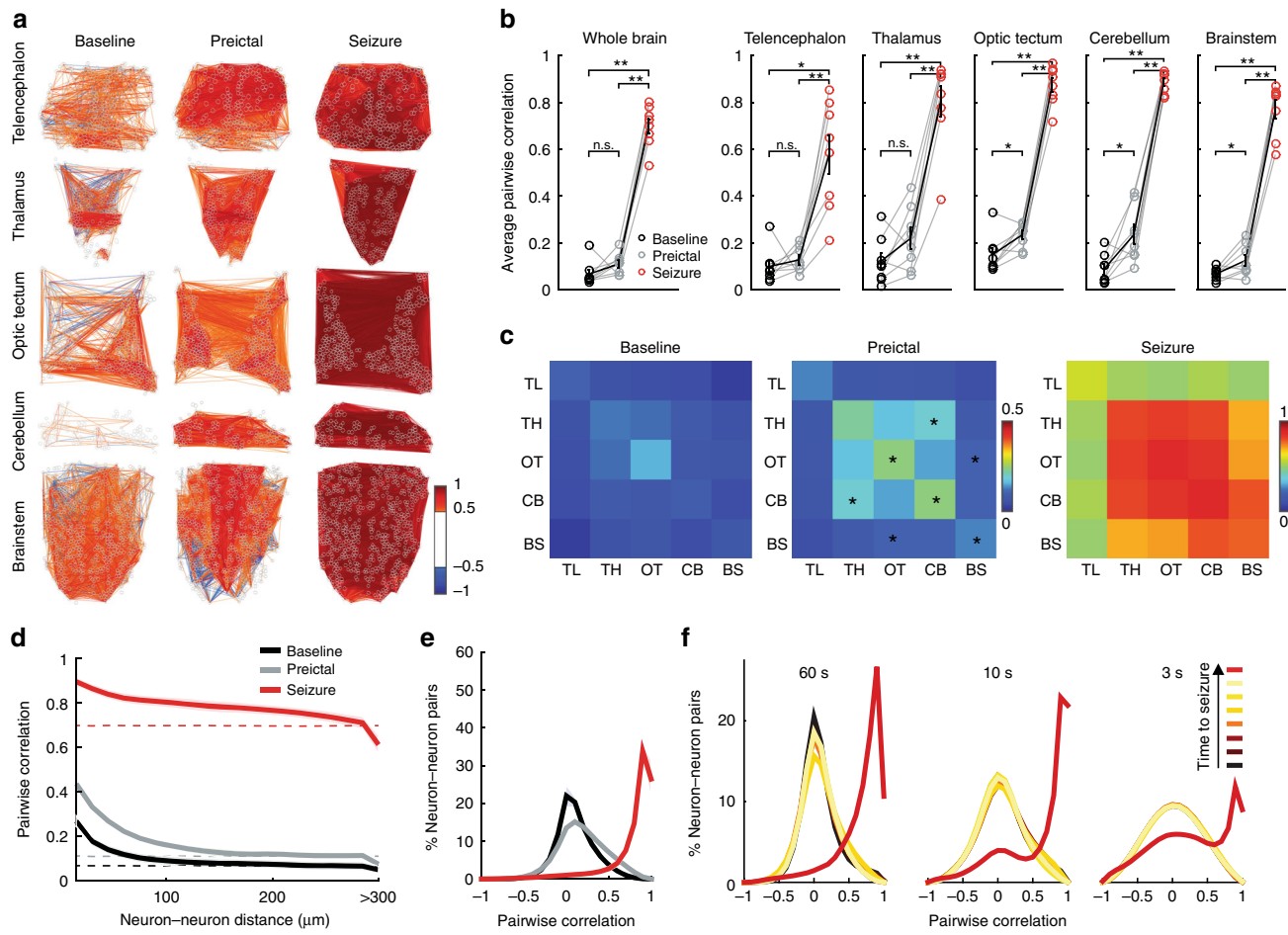

**Fig. 3** Functional connectivity between neurons changes abruptly from the preictal period to the generalized seizure. **a** A map depicting pairwise Pearson's correlation coefficients of neural activity within brain regions during baseline, preictal and seizure periods. Each colored line indicates a strong positively (>0.5 in red) or negatively (<−0.5 in blue) correlated activity between pairs of neurons located at the end of the lines. **b** Average pairwise Pearson's correlation during baseline (black), preictal (gray), and seizure (red) periods across the whole brain, and within individual brain regions. **c** Correlation matrices indicating average pairwise Pearson's correlations of neural activity during baseline, preictal and seizure, across brain regions (telencephalon, TL; thalamus, TH; optic tectum, OT; cerebellum, CB; and brainstem, BS). Warmer colors indicate stronger positive correlations. **d** Relation between pairwise correlation of neural activity and distance between each neuron pair. Dotted lines represent the results when neural locations are shuffled. **e** Histogram representing the distribution of all correlation coefficients between neurons from all animals during baseline (black), preictal (gray), and seizure periods (red). **f** Correlation coefficients during seven time periods immediately preceding a generalized seizure. The time periods are of 60, 10, and 3 s length, respectively. Lighter colors indicate temporal proximity to the seizure. $^{**}p = < 0.01$, $^*p = < 0.05$, ns = not significant, Wilcoxon signed-rank test. Error bars (**b**) and shaded regions (**d**–**f**) represent the s.e.m. of $n = 8$ fish

independent of spatial constraints associated with neural connectivity rules. Epileptic seizures induced by pilocarpine, a muscarinic acetylcholine agonist, exhibit similar features, but the ictogenesis occurs less rapidly[34] (Supplementary Figs. 1 and 4, Supplementary Movie 3).

**Glial cells are active and synchronized during preictal period.** The rapid and non-spatially constrained transition of neural activity and connectivity during seizure generation suggests that the underlying mechanisms are beyond neural connectivity rules mediated by classical synaptic transmission. These findings further highlight a potential role for non-neuronal cells in the brain, such as glia, for the spread of seizures. To study glial activity during seizure development, we generated a *Tg(GFAP:Gal4)nw7* zebrafish line expressing *Gal4* under the astrocytic *glial fibrillary acidic protein (GFAP)* promoter[35,36]. In zebrafish, *GFAP* promoter was shown to label radial glial cells, which in addition to

their neurogenic capacity[35], serve the function of mammalian astrocytes[37,38]. We first demonstrated that *GFAP:Gal4* expressing cells are primarily glia and not neurons. To this end, we crossed *Tg(GFAP:Gal4)nw7;Tg(UAS:GCaMP6s)* fish with *Tg(elavl3:jRCaMP1a)*[39] fish expressing the red-shifted calcium indicator *jRCaMP1a* in all neurons. Confocal images revealed exclusively *GFAP* expressing cells along the ventricular zones at the midline, whereas we observed more overlap with *elavl3* positive neurons in other parenchymal zones (Fig. 4a). To solely study the activity of *GFAP* expressing glial cells, we focused our analysis on the region along the ventricles[40] with no neuronal labelling. Next, we performed calcium imaging on *Tg(GFAP:Gal4)nw7;Tg(UAS:GCaMP6s)* zebrafish, where PTZ was added after a baseline period, as described earlier with *Tg(elavl3:GCaMP6s)* fish. We observed strong increase in glial calcium signals after the perfusion of PTZ (Fig. 4b, and Supplementary Movie 2) that was distinct from the neural activity we recorded (Fig. 2b). Simultaneous patch-clamp recordings of individual glia and imaging of

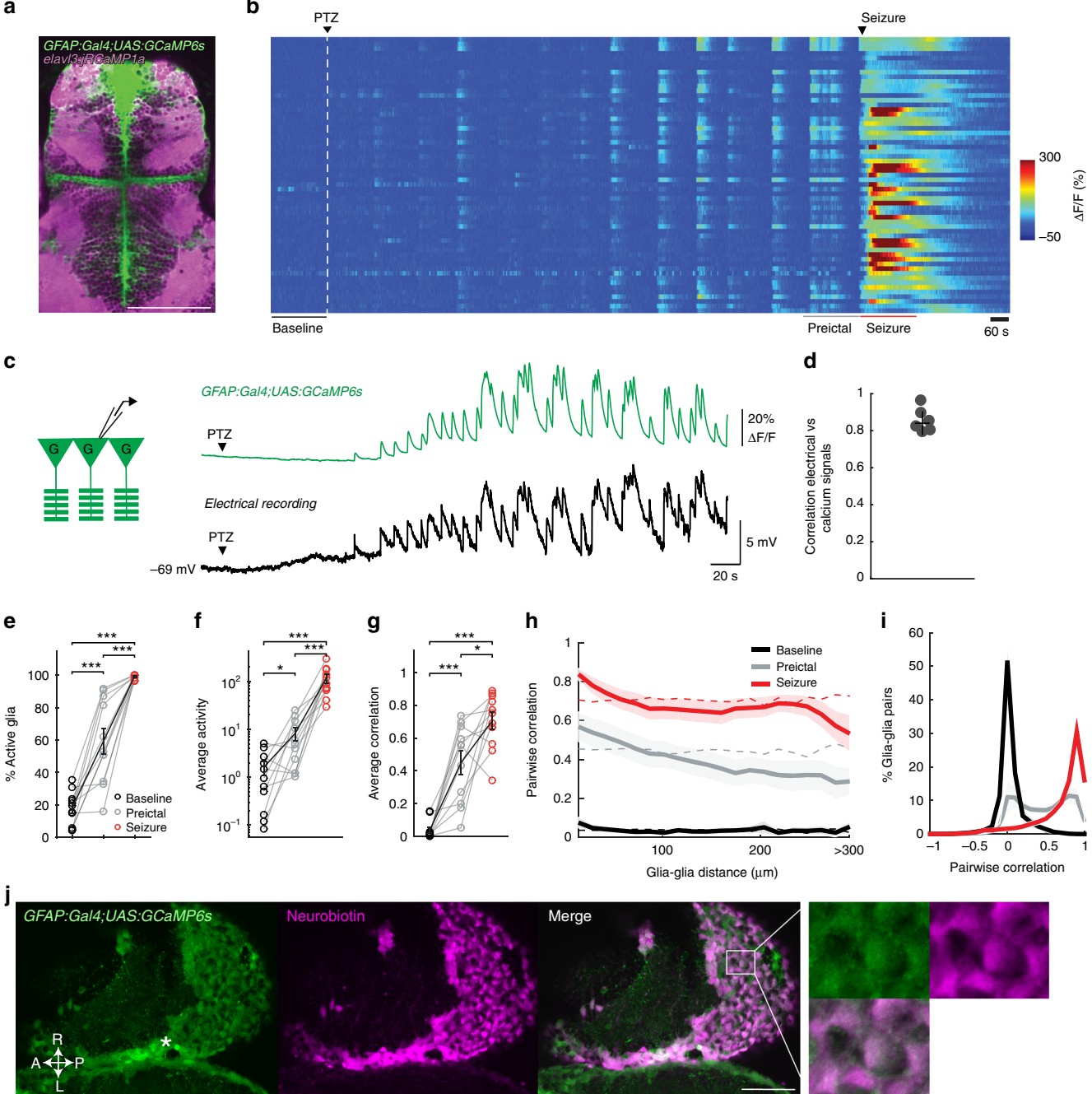

**Fig. 4** Glial cells are highly active and strongly synchronized already during preictal period. **a** Confocal image of a transgenic zebrafish larvae expressing *GCaMP6s* in *GFAP* positive glial cells (green) and *jRCaMP1a* in *elavl3* positive neurons (magenta), dorsal view. **b** Glial calcium signals measured by two-photon microscopy in *Tg(GFAP:Gal4)nw7;Tg(UAS:GCaMP6s)* transgenic zebrafish larva. White dashed line indicates the start of 20 mM pentylenetetrazole (PTZ) perfusion. **c** Simultaneous patch-clamp recording of glial membrane potential and epifluorescence calcium recordings of *Tg(GFAP:Gal4)nw7;Tg(UAS:GCaMP6s)* transgenic zebrafish, during PTZ-induced epileptic activity. **d** Correlation of electrical activity and calcium signals from individual glial cells, $n = 6$ cells/zebrafish. **e** Percentage of active glial cells ($>3\mathrm{std}_{baseline}$) during baseline (black), preictal (gray), and seizure (red) periods. **f** Average activity of the active glial cells, defined by the area under the curve of the $\Delta F/F$ trace. **g** Average pairwise Pearson's correlation between glial cells. **h** Relation between pairwise correlation of glial activity and the distance between each glial pair. Dotted lines represent the results when glial locations are shuffled.
**i** Histogram representing the distribution of the correlation coefficients between glial cells during baseline (black), preictal (gray) and seizure periods (red).
**j** Confocal image showing *GFAP* positive radial glia expressing *GCaMP6s* (green) and neurobiotin coupling between glial cells (magenta) after filling a single glia with neurobiotin by patch-clamp electrode. Location of the patch-clamped glial cell is indicated with *. A, anterior; P, posterior; L, left; R, right. White bars reflect 100 µm (**a**, **j**). ***$p = < 0.001$, *$p = < 0.05$, ns = not significant, Wilcoxon signed-rank test. Error bars and shaded regions represent the s.d. of $n = 6$ cells (**d**) or s.e.m. of $n = 11$ fish (**e**–**i**)

calcium signals revealed strong depolarization of glial membrane potential that was highly correlated with the glial calcium signals, during PTZ-induced epileptic activity (Fig. 4c, d). After the addition of PTZ, we noticed large, long lasting and highly synchronous glial calcium events in the preictal period, as well as very large calcium increase during generalized seizures (Fig. 4b, and Supplementary Movie 2). Upon quantification of glial calcium signals, we measured significant increases in the percentage of active glial cells (Fig. 4e) and in the average activity of glial cells (Fig. 4f). During the preictal period, contrary to neurons, glial cells displayed a large and significant increase in synchrony (Fig. 4g–i) that spreads across large distances (Fig. 4h). Interestingly, we observed that glial cells in the telencephalon were more active and more correlated than those in the thalamus, during the preictal period (Supplementary Fig. 5). During generalized seizures, all glial cells were highly synchronized (Fig. 4g–i) across the brain (Fig. 4h), similar to neurons (Fig. 3d). Importantly, we observed similar glial activity and synchrony during pilocarpine-induced seizures (Supplementary Fig. 7, Supplementary Movie 4).

Strong synchrony within glial networks as we observed across the zebrafish brain, suggests the presence of gap junctions between glial cells, similar to astrocytic gap junctions in mammals. In fact, we observed strong neurobiotin dye coupling between glial cells across large zebrafish brain areas, when individual glial cells were filled with neurobiotin during patch-clamp recordings (Fig. 4j). We also detected high levels of the transcripts for the gap junction protein connexin 43 in zebrafish glial cells[41] (Supplementary Fig. 6). Taken together, our results show that the gap junction coupled glial cells are highly active and synchronized during the preictal period, highlighting an important difference between the glial and neural networks preceding generalized seizures.

**Glia-neuron interactions change drastically during seizures**. Our analyses of glial activity suggest an important role for glia during the transition of preictal activity to a generalized seizure. To investigate the functional interactions between glia and neurons, we measured glial and neural calcium signals simultaneously. To achieve labelling of both glia and neurons, we injected Et(-0.6hsp70l:Gal4-VP16)s1020t;Tg(UAS:GCaMP6s) zebrafish expressing GCaMP6s in only thalamic neurons with GFAP:Gal4 plasmid at the one-cell stage. This approach allowed us to visualize the calcium signals of both thalamic neurons and sparsely labelled GFAP expressing glial cells along the ventricles (Fig. 5a). Consistent with our previous data (Figs. 3 and 4), we observed a strong and synchronous glial activity before and during generalized seizures, while the neural activity and synchrony was strongly increased only during generalized seizures (Fig. 5b–d). To investigate functional interactions between glia and neurons we next calculated pairwise Pearson's correlations between every glia and neuron. Interestingly, we observed that the synchronization of glial and neural activity drastically increased only during generalized seizures (Fig. 5d).

To examine the temporal relationship between glial and neural activity further, we detected each burst of glial activity and plotted average activity of all simultaneously imaged glia and neurons (Fig. 5e). Interestingly, we observed that during the preictal period neural activity bursts came first and were followed by glial activity bursts that coincided with the reduction of neural activity. As a result, bursts of neural and glial activity were anticorrelated, which highlights a reduction of neural activity during the period of glial bursts. This picture, however, altered completely once the first generalized seizure was initiated and during the following bursts of generalized seizures. We found that during the first generalized seizure neural activity preceded the glial activity, but

the neural activity grew drastically as soon as the glial activity was initiated. During the period of continuous generalized seizures, glial and neural activity was initiated simultaneously. We infer that the preictal glial activity that corresponds to dampening of neural activity might reflect a protective function of glial network, preventing the spreading of small preictal neural activity bursts. However, during generalized seizures, the glial activity overlaps with large neural activity and might be related to an abrupt release of glutamate in the brain through glial cells. To test this hypothesis, we imaged the glutamate levels near glial cells by using transgenic glutamate sensor iGluSnFR[42], expressed specifically in glia under GFAP promoter in Tg(GFAP:iGluSnFR) transgenic zebrafish[43]. We observed no change in extracellular glutamate levels near glial cells until the generalized seizures, when transient increase of extracellular glutamate occurred (Fig. 5f). All these results highlight that glia-neuron interactions change during the course of seizure generation. During the preictal period, increased glial activity bursts follow neural activity with a delay and corresponds to the dampening of neural activity. Whereas during generalized seizures, we observed a strong functional and temporal coupling between glial and neural networks as well as a large surge in extracellular glutamate near glia, emphasizing a potential role for glia-neuron interactions and massive glutamate release in the spreading of ictal activity across the brain.

**Activation of glia excites nearby neurons**. One potential mechanism for synchronized glial activity to trigger a generalized seizure might be through a direct excitation of neural networks. Such an excitation by glia onto neurons could in principle be delivered across large distances through gap junction coupled glial networks (Fig. 4j, Supplementary Fig. 6), which can explain why neural networks do not follow functional connectivity rules during generalized seizures. To test whether the activation of glial cells can indeed trigger significant excitation of nearby neurons, we expressed channelrhodopsin-2 (ChR2) in glial cells using Tg(GFAP:Gal4)nw7;Tg(UAS:ChR2-mCherry) zebrafish line[44]. Optogenetic activation of glial cells upon blue light stimulation led to a sharp depolarization of glial membrane potential (Fig. 6a). We observed neurobiotin coupling from the individual electrophysiologically recorded glia towards large populations of glial cells, even beyond the glial patches that sparsely expressed ChR2-mCherry (Fig. 6b). Optogenetic activation of glial cells led to significant excitation of nearby neurons, but not those neurons that are far from glial patches expressing ChR2 (Fig. 6c–g). Interestingly, we observed that neurons with strong excitation amplitude usually showed very short delay, whereas neurons with small but significant excitation showed rather long excitation onset delays, upon glial activation (Fig. 6h). To further investigate the nature of glia-neuron connections, we tested the joint effect of ionotropic glutamate receptor blockers NBQX and AP5. The glia-neuron communication strength was significantly reduced in the presence of NBQX/AP5, especially in those neurons with medium connection strength and slow kinetics (Fig. 6j, k). We also observed that few glia-neuron connections still showed significant amount of excitation even in the presence of NBQX/AP5 (Fig. 6i, k). Inspired with these results, as well as the gap junction coupling between glial cells, we tested the effect of gap junction blocker carbenoxolone in glia-neuron communication. We observed that the delivery of carbenoxolone reduced the strength of neural excitation, upon optogenetic activation of ChR2 expressing glial patches (Fig. 6l–n). Taken together, we show that glial activation can strongly modulate the activity of neurons by delivering strong and transient excitation. This strong modulation of neural activity by glia through the action of both glutamate and gap junctions

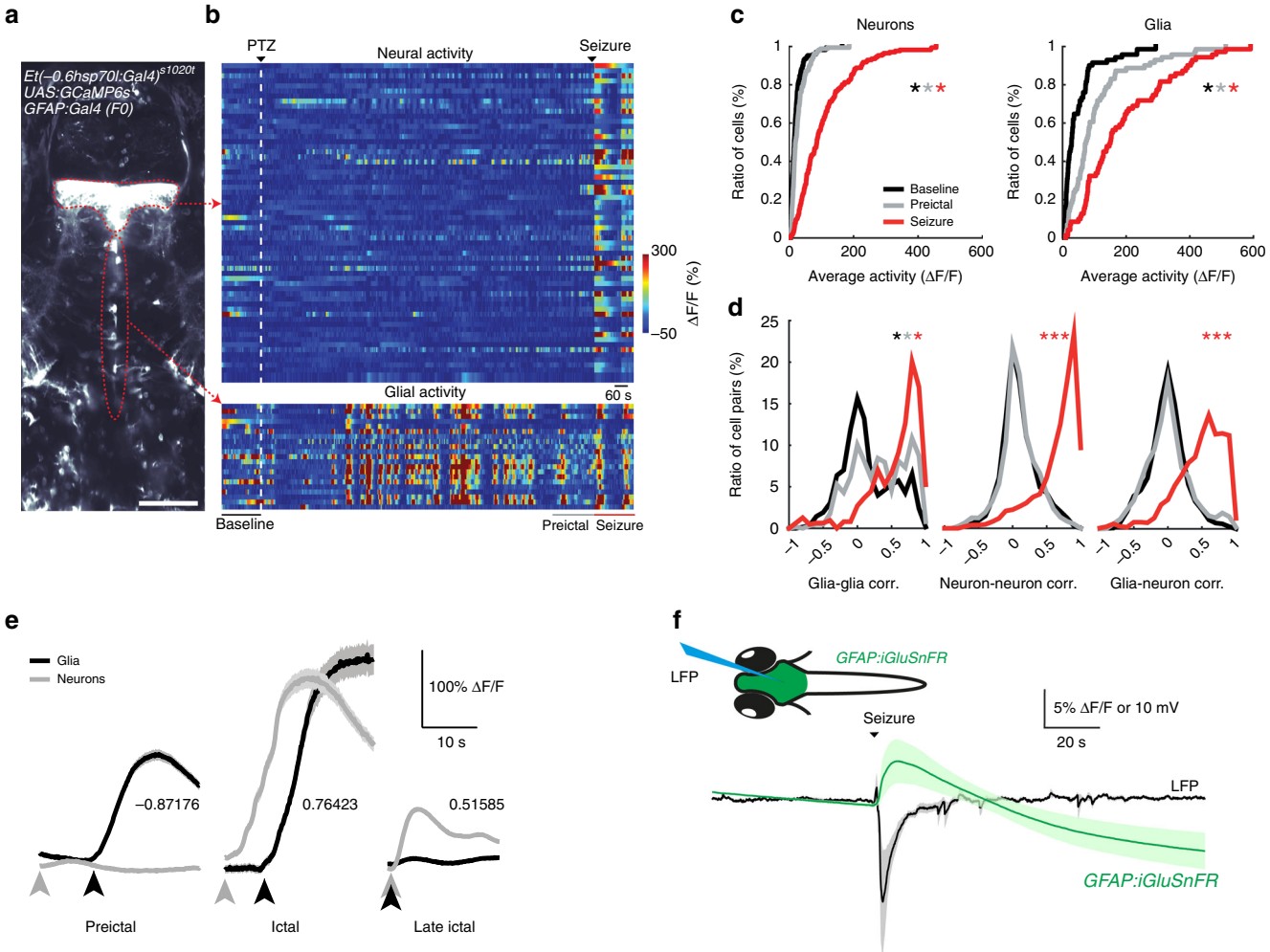

**Fig. 5** Functional interactions between glia and neurons change drastically during seizure generation. **a** An optical section of a transgenic zebrafish larva expressing *GCaMP6s* in thalamic neurons (top red dotted line) and *GFAP* expressing glial cells near the ventricles (bottom red dotted line) obtained by two-photon microscopy, dorsal view. White bar reflects 100 μm. **b** Activity (ΔF/F) of individual thalamic neurons (top) and glial cells along the ventricle (bottom). White dashed line indicates the start of 20 mM pentylenetetrazole (PTZ) perfusion. Warmer colors indicate stronger activity. **c** Cumulative distribution of neural (left) and glial activity (right) during baseline (black), preictal (gray) and seizure (red) periods, $n = 71$ glial cells and $n = 171$ neurons across four fish. **d** Histograms representing the distribution of all pairwise Pearson's correlations for the activity of glia-glia pairs (left), neuron-neuron pairs (middle) and glia-neuron pairs (right). **e** Temporal relationship between average activity bursts of glia (black) and neurons (gray) during preictal and ictal state. Average Pearson's correlation values between glial and neural activity bursts are indicated. **f** Scheme representing simultaneous local field potential (LFP) recording and epifluorescence imaging of a transgenic zebrafish larva expressing the glutamate sensor *iGluSnFR* (upper). The lower graph shows the average local field potential signals (black), and the average fluorescence intensity change (ΔF/F) for *iGluSnFR* signals (green). Signals from multiple fish aligned at the onset of generalized seizure, $n = 7$ fish. ***$p = < 0.001$, *$p = < 0.05$, ns = not significant, Wilcoxon rank-sum test. Shaded regions denote se. m. of $n = 4$ fish (**d, e**) and $n = 7$ fish (**f**)

could underlie the coupling between glial and neural networks during generalized seizures.

## Discussion

The transition from a preictal state to a generalized seizure is an abrupt process accompanied by strong alteration of the functional connectivity between glial and neural networks. Thanks to the small size of the zebrafish brain, we could follow the seizure development in thousands of individual neurons and glia across multiple brain regions. Such a combination of scale and detail would be difficult in studies of epilepsy in mammals. Using simultaneous electrical recordings, we confirmed that genetically encoded calcium indicators in combination with two-photon imaging can reliably report epileptic activity. Hence, we propose that two-photon calcium imaging is complementary to

commonly used high-throughput behavioral assays in zebrafish for studying mechanisms underlying seizure generation[24,25].

We provide evidence for several common physiological features between zebrafish and mammalian models of epilepsy. Previous studies suggest that the seizure propagation arises from profound alterations in local and global network connectivity[2,45]. In our seizure models, we observed recruitment of more neurons as the network shifts from baseline to preictal state. This was accompanied by a small and brain-region-specific increase of neural synchrony between nearby neurons. Consistent with previous reports[4], we observed that this enhanced preictal neural activity and synchrony was different across brain regions, probably due to differences in the distribution of excitatory and inhibitory neural populations. Interestingly, among the five major brain regions we monitored, the optic tectum and cerebellum were the brain regions which showed the most prominent

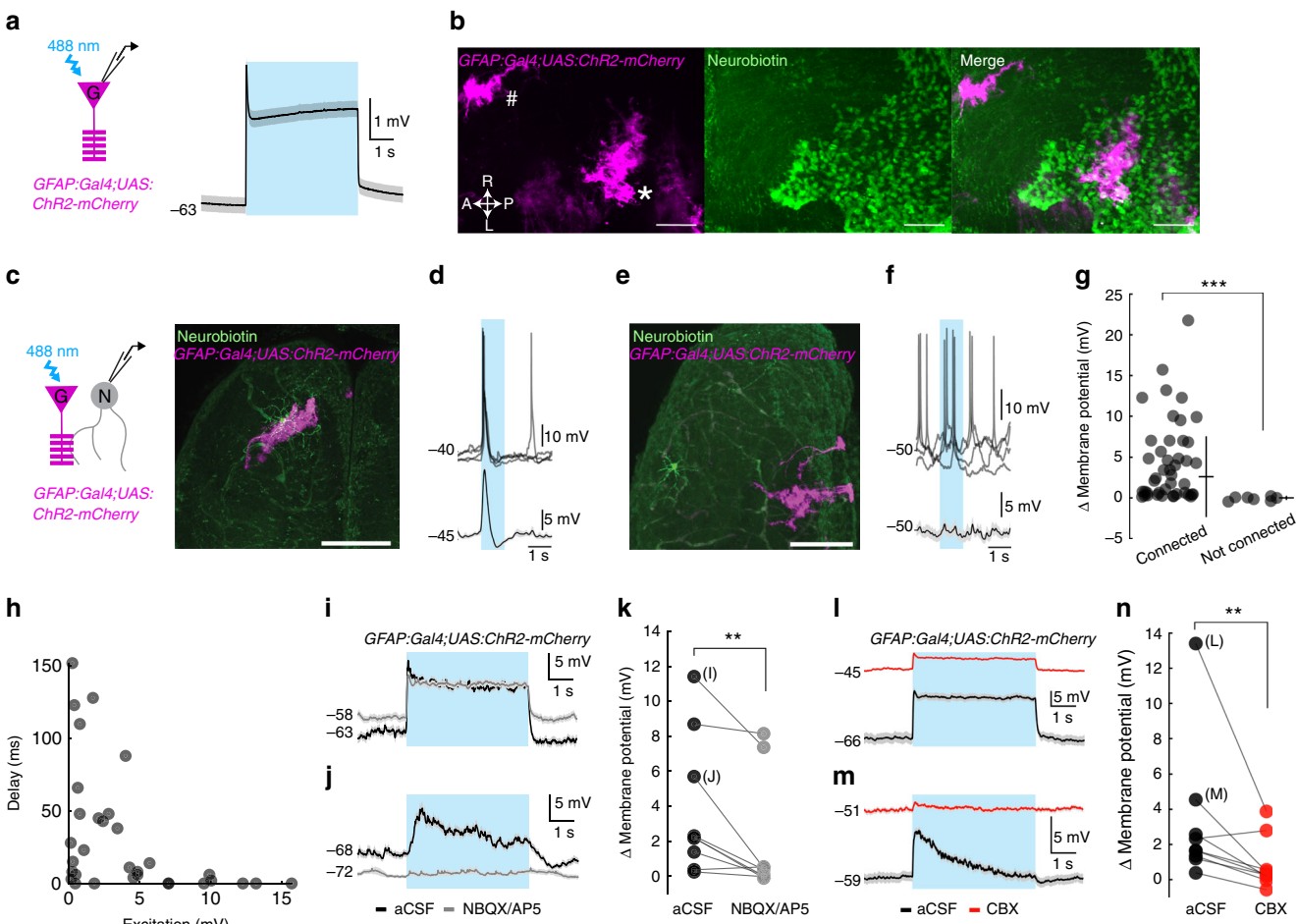

**Fig. 6** Optogenetic activation of glial cells excites nearby neurons. **a** Scheme for patch-clamp recording of glial cells expressing *channelrhodopsin2* (left). Average membrane potential of a representative optogenetically activated glia (right), *n* = 3 glia. **b** Zebrafish brain sparsely expressing *channelrhodopsin2-mCherry* in *GFAP* positive glial cells (magenta, indicated with * and #). A single glial cell filled with neurobiotin (green) using patch-clamp electrode, at location *. Note the neurobiotin coupling across glia. **c**, **e** Dorsal view of brain-explant preparation showing sparse expression of *channelrhodopsin2* in *GFAP* positive glial cells (magenta), and individual neurons filled with neurobiotin (green) during intracellular recordings. **d**, **f** Membrane potential of example neurons with processes nearby *channelrhodopsin2* expressing glial cells (**c**, **d**), or far away from *channelrhodopsin2* expressing glial cells (**e**, **f**) during 1 s optogenetic stimulation (blue shade). Overlaid sweeps (top) and 30 Hz low-pass filtered and averaged membrane potential (bottom). **g** Average change in membrane potential of neurons during the first 300 milliseconds of optogenetic stimulation of glial cells. *n* = 43 connected, *n* = 7 not connected neurons. **h** Distribution of excitation amplitude and excitation delay of connected neurons, upon optogenetic activation of glial cells. *n* = 43 connected neurons. **i**, **j** Examples of neurons with fast (**i**) and slow (**j**) excitation upon optogenetic activation before (black) and after (gray) addition of ionotropic glutamate receptor blockers NBQX (5 μM) and AP5 (25 μM). **k** Average change in membrane potential of neurons during first 1 sec of optogenetic stimulation of glial cells, before and after the addition of NBQX and AP5, *n* = 9 neurons. **l**, **m** Examples of neurons with fast (**l**) and slow (**m**) excitation upon optogenetic activation before (black) and after (red) addition of gap junction blocker carbenoxolone (CBX, 100 μM). **n** Average change in membrane potential of neurons during the first 1 s of optogenetic stimulation of glial cells, before and after addition of carbenoxolone, *n* = 9 neurons. **p* = < 0.01, ***p* = < 0.001, Wilcoxon ranksum (**g**) or sign-rank test (**k**, **n**). Shaded regions denote s.e.m. of 25 repetitions (**a**, **d**, **f**, **i**, **j**, **l**, **m**), error bars denote mean ± s.d. (**g**). White bars reflect 50 μm (**b**) or 100 μm (**c**, **e**)

alterations during the preictal period, followed by the thalamus and brainstem. On the contrary, the zebrafish telencephalon, homologous to mammalian cortex[46], was recruited only during generalized seizures and always with a significant delay. It was proposed that the spreading of epileptic activity may require involvement of epileptogenic hubs[4]. As such, these confined areas, which have profound connections to the rest of the global network would allow oscillatory activity to spread from local to global networks, resulting in generalized seizures. Similar to the mammalian brain, our results suggest that the zebrafish midbrain regions such as thalamus and optic tectum (homologous to mammalian superior colliculus) appear to have hub-like properties with earlier recruitment of synchronous activity before generalized seizures. These findings highlight a potential role of

non-cortical brain regions in the initiation and spreading of seizures. Interestingly, the first epilepsy with a proven monogenic etiology, autosomal dominant sleep-related hypermotor epilepsy (ADSHE), is accompanied by prominent alterations in these midbrain regions in human patients[47]. Future studies on the role of non-cortical brain regions in human epilepsies will likely provide a better understanding of underlying mechanisms.

The abrupt transition from a preictal state to a generalized seizure was accompanied by a drastic reorganization of functional connectivity and synchrony across neurons. Similar to epilepsy patients and rodent models[11,12], our electrical LFP recordings in zebrafish brain suggest a reduction of high-frequency activity and an increase in low-frequency activity during seizure generation. We observed that the emerging hypersynchronous connectivity

during the generalized seizure is independent of spatial constraints of baseline or preictal connectivity state in two very different pharmacological seizure models; PTZ and pilocarpine. Furthermore, functional connectivity patterns during generalized seizures exhibit no or little negative correlations that would be driven by inhibitory connections. All these drastic alterations in brain activity and connectivity during seizure propagation is in line with the general view of epileptic seizures as state transitions, where brain networks shift from a balanced state to a hypersynchronous or hyperconnected state[13,45]. Intriguingly, epileptic seizures in human patients are more likely to occur during transitions of sleep states which show profound alterations in brain synchrony and connectivity[48]. Similar to epileptic state transitions, higher frequency desynchronized activity during wakefulness and rapid eye movement (REM) sleep is replaced by low-frequency oscillations during slow wave sleep[49]. Moreover, thalamocortical and corticocortical pathways that are involved in sleep[50] can facilitate long range connectivity and are proposed to play important roles in epileptic seizures[4,10]. Hence, investigating parallels between sleep state transitions and epileptic state transitions are promising avenues for better understanding of common underlying mechanisms.

The rapid alterations of neural activity and connectivity that we observed during the transition from preictal to ictal period, was difficult to explain by only considering the rules of neural connectivity mediated by classical synaptic transmission. The abrupt and excessive increase in positive correlations that is not constrained by spatial distances between neurons encouraged us to investigate the properties of glial cells during seizure generation. Our results showed a direct involvement of glial networks during seizure generation in two different pharmacologically induced zebrafish seizure models and therefore are likely to have broad relevance. In both of these seizure models, we observed consistent behaviors of glial and neural networks. Already during the preictal phase, glial networks were highly active and synchronized across large distances, when compared to rather small and local increase of neural activity and synchrony. Strong synchronization of glial networks across large distances are likely due to gap junction coupling between glial networks, of which we provide direct evidence by neurobiotin coupling between glia cells and glia specific expression of connexin 43. These findings are in line with the idea that glial networks are strongly coupled and can act as a single unified entity with highly coordinated activation[51]. Interestingly, during the preictal phase we observed that glial activity bursts coincide with the dampening of neural bursts and is anticorrelated with neural activity. Yet, this picture changed completely during generalized seizures, which are characterized by an abrupt increase in correlated activity between glial and neural networks. Altogether, our results suggest that the initiation and spreading of generalized seizures can be mediated by a transient alteration of glia-neuron interactions.

Astrocytes modulate synaptic transmission and neural excitability[15,52], and several mutations in glia associated genes were linked with epilepsy[21,53]. Especially, gap junction coupling between astrocytes was shown to play important roles in redistribution of ions and neurotransmitters across large distances in the brain[15,16,19,20,51,52]. Consistent with protective functions of glia, we observed that preictal glial activity appears to dampen neural activity bursts. We propose that during this preictal state, large synchronous glial activity represents the homeostatic function of glia that can absorb excessive levels of glutamate and cations through the major glial excitatory amino acid transporter, EAAT2 (also known as glutamate transporter 1, GLT1; and solute carrier family 1 member 2, SLC1A2)[54], and redistribute these substances across the gap junction coupled glial networks[55]. In accordance with this hypothesis, GLT1 knockout mice display

spontaneous seizures[56] and mutations in SLC1A2 gene are associated with epilepsy in human patients[53]. Therefore, we propose that preictal glial activity in our recordings reflects the protective function of glia against epileptic seizures. Interestingly, it was reported that gap junction coupling between astrocytes is diminished during epileptogenesis and results in a reduced homeostatic potential of astrocytes[21,22,51]. Moreover, deficiency in the astrocytic gap junction coupling was shown to cause temporal lobe epilepsy[21,22].

What can lead to an abrupt alteration in glia-neuron interactions from preictal to ictal state? We propose that prolonged preictal activity fatigues the homeostatic function of glia. The cellular mechanisms for an eventual collapse of the homeostasis and the precise trigger of the generalized seizures are yet to be identified. A potential mechanism for such rapid alterations in glial function might be directly related to the way the major glial glutamate transporter, EAAT2, operates[56]. Extracellular glutamate anions accompanied by $Na^+$ and $H^+$ are exchanged with intracellular $K^+$ ions[55,57,58]. Hence, the clearing of excessive glutamate in the preictal brain is a process heavily dependent on electrochemical gradients. It was reported that increased potassium concentrations around glial cells may reverse the uptake leading to a release of glutamate into the extracellular space[55,59]. It is likely that during preictal activity extracellular potassium levels reach a threshold, which prevents EAAT2 from functioning. The glial network may eventually collapse, triggering a massive release of glutamate from the strongly gap junction coupled glial network, initiating a generalized seizure. In fact, our optogenetic activation of small number of glial cells in zebrafish brain was effective in generating transient and strong excitation of nearby neurons through the action of glutamate and gap junctions, highlighting a potential way to excite neurons through gliotransmission[17,18]. In line with these ideas, we observed a large rise of extracellular glutamate near glial cells only during generalized seizures. Further experiments will be important for a better mechanistic understanding of the role of glial networks both in prevention and induction of epileptic seizures.

While in this study we focused on epileptic seizures, glia-neuron interactions are likely to be involved in many other types of state transitions within the brain networks. Glia-neuron interactions are linked to a variety of neurological diseases. Increased expression of glial connexin 43 was shown in humans and animal models of neurodegenerative disorders like Alzheimer's disease and Parkinson's disease[60,61]. Both glial and neuronal connexins have been shown to contribute to high-frequency gamma oscillations[62,63], which are proposed to be important for cognitive functions. Furthermore, impairment of astrocytic potassium handling may lead to enhanced network excitability and increase in high-frequency bands[64]. Interestingly, a reduction in gamma activity was reported in Alzheimer's disease, while an increased activity was shown in epilepsy[65]. Moreover, an expanding number of studies propose diverse roles for glia-neuron interactions in brain functions such as regulating locomotion, motivation, memory formation, and sleep[66–68]. Given the large number of glial cells in the brain, their prominent role in regulating neural excitability and activity is not that surprising. Future studies on glia-neuron communication will shed further light on the role of glia-neuron interactions in the physiology and pathophysiology of brain circuits.

## Methods
**Contact for reagent and resource sharing**. Details of all key reagents and resources used in this article are included in Supplementary Tables 1 and 2. Further information and requests for reagents may be directed to and will be fulfilled by the lead author Emre Yaksi (emre.yaksi@ntnu.no).

**Experimental model and subject details**. Fish maintenance: fish were kept in 3.5-liter tanks at a density of 15–20 fish per tank in a Techniplast Zebtech Multilinking system at constant conditions: 28 °C, pH 7, 6.0 ppm $O_2$ and 700 µS, at a 14:10 h light/dark cycle to simulate optimal natural breeding conditions. Fish received a normal diet of dry food (Zebrafeed, Sparos I&D Nutrition in Aquaculture, <100–600, according to their size) two times per day and *Artemia nauplii* once a day (Grade0, platinum Label, Argent Laboratories, Redmond, USA). Larvae were maintained in egg water (1.2 g marine salt, 20 L RO water, 1:1000 0.1% methylene blue) from fertilization to 3 dpf and in artificial fish water (AFW, 1.2 g marine salt in 20 L RO water) from 3 to 5 dpf. The animal facilities and maintenance of the zebrafish, *Danio rerio*, were approved by the Norwegian Food Safety Authority (NFSA) and Belgian government. All experimental procedures performed on zebrafish larvae up to 5 days post fertilization were in accordance with the directive 2010/63/EU of the European Parliament and the Council of the European Union and the Norwegian Food Safety Authorities. Experimental procedures performed on zebrafish larvae older than 5 dpf were further approved by the Ethical Committee of KULeuven in Belgium (P088-2014) and Norwegian Food Safety Authority.

For experiments, the following fish lines were used: *Tg(elavl3:GCaMP6s)*[28], *Tg (elavl3:jRCaMP1a)*[39], *Tg(UAS:ChR2-mCherry)*[44,69], *Tg(gad1:GFP)*[32], *Tg(vglut2a: dsRED)*[33], *Et(-0.6hsp70l:Gal4-VP16)s1020t*[70], *Tg(UAS:GCaMP6s)*[71], *Tg(GFAP: iGluSnFR)*[43]. *Tg(GFAP:Gal4)nw7* transgenic animals were generated in our lab upon coinjection of tol2 transposase mRNA and *GFAP:Gal4* plasmid obtained from the Ohshima lab[35]. The *Tg(GFAP:Gal4)nw7* expression pattern was identified in three independent founders.

Combined LFP experiments together with epifluorescence calcium and iGluSnFR imaging: Simultaneous epifluorescence calcium imaging or iGluSnFr imaging was performed together with LFP recordings of seizure activity on 7 days old *Tg(elavl3:GCaMP6s)* ($n = 5$) and 5 days old *Tg(GFAP:iGluSnFR)* zebrafish ($n = 7$). Zebrafish larvae were first anesthetized with 0.02% MS222 and paralyzed by α-bungarotoxin injection. They were embedded in 1–1.5% low melting point agarose (LMP, Fisher Scientific) in a recording chamber (Fluorodish, World Precision Instruments). The layer of LMP agarose covering the dorsal side of the forebrain was carefully removed, so it was accessible for electrode placement. The fish were placed under the epifluorescence microscope (Olympus BX51 fluorescence microscope, Olympus Corporation). AFW was constantly perfused during the experiments. For electrophysiological recordings, a borosilicate glass patch clamp pipette (9–15 MOhms) loaded with teleost artificial cerebrospinal fluid[72] (ACSF, containing in mM: 123.9 NaCl, 22 D-glucose, 2 KCl, 1.6 MgSO$_4$ · 7H$_2$O, 1.3 KH$_2$PO$_4$, 24 NaHCO$_3$, 2 CaCl$_2$ · 2H$_2$O) was inserted into the optic tectum. Electrical recordings were performed in current clamp mode with a high impedance amplifier and band pass filtered at 0.1–1000 Hz, at a sampling rate of 10 KHz (MultiClamp 700B amplifier, Axon instruments, USA). Calcium signals were recorded using an EMCCD camera (Hamamatsu Photonics) at sampling rate of 25 Hz. iGluSnFR signals were recorded using an Allied Vision Manta Camera. After electrode placement the larvae were left 10 min for stabilization to ensure fish were fully awake and recovered from the anesthesia, MS222. Baseline activity was recorded for 10 min, then 20 mM pentylenetetrazole (PTZ) dissolved in AFW was delivered via perfusion system until the end of the experiment. Duration of the total recording was 90 min. Data acquisition of both signals (LFP and calcium) was performed with a custom code written in MATLAB (Mathworks).

Confocal anatomical imaging: For confocal anatomical imaging, 5 days old *Tg (GFAP:Gal4)nw7;Tg(elavl3:jRCaMP1a)* and *Tg(gad1:GFP);Tg(vglut2a:dsRED)* zebrafish larvae were anesthetized with 0.02% MS222 and embedded in 1.5–2% LMP agarose. Anatomical Z-scans were acquired using a Zeiss Examiner Z1 confocal microscope with a ×20 plan NA 0.8 objective, using ×10 average for each plane.

Combined optogenetical stimulation and electrophysiological recordings: For combined optogenetical stimulation and electrophysiological recordings, 4–5-weeks-old zebrafish were anesthetized in ice cold water and sacrificed by decapitation. Their brains were dissected out in cooled and oxygenated ACSF. For intracellular recordings, borosilicate glass capillaries of 9–15 MOhms were filled with intracellular solution which contained (in mM): 130 κ-gluconate, 10 Na-gluconate, 10 HEPES, 10 Na$^{2+}$-Phospho-Creatine, 4 NaCl, 4 ATP-Mg and 0.3 Na$^3$ $^+$-GTP. Electrical signals were recorded by MultiClamp 700B amplifier at sampling rate of 10 KHz. All recordings and data analyses were performed using custom codes written in MATLAB. Activation of *channelrhodopsin2* was done by flashing a 480 nm LED light, through the optical path of the Olympus BX51 microscope for a duration of 1 sec. For morphological reconstructions, neurobiotin (0.5%, Vectorlabs) was added to the intracellular solution. Brains were fixed overnight in 4% PFA at 4 °C, and afterwards incubated first with PBS and later with 5% streptavidin (Vectorlabs) in 0,5% PBSTx. Brains were imaged using a Zeiss Examiner Z1 confocal microscope. To block ionotropic glutamate receptors, we used 5 µM NBQX (Tocris) and 25 µM D-AP5 (Tocris). To block gap junctions, we used 100 µM carbenoxolone (Tocris).

Two-photon calcium imaging: Two-photon calcium imaging was performed on 7 days old *Tg(elavl3:GCaMP6s)* ($n = 8$, PTZ), 5 days old *Tg(elavl3:GCaMP6s)* ($n = 6$, pilocarpine), 5 days old *Et(-0.6hsp70l:Gal4-VP16)s1020t;Tg(UAS:GCaMP6s)* ($n = 4$) and 5 days old *Tg(GFAP:Gal4)nw7;Tg(UAS:GCaMP6s)* zebrafish larvae ($n = 11$, PTZ and $n = 7$, pilocarpine). Animals were paralyzed with α-bungarotoxin (Invitrogen BI601, 1 mg/mL) and embedded in 1.5–2% LMP agarose in a recording chamber (Fluorodish, World Precision Instruments). The recordings were performed in two-photon microscopes (from Thorlabs Inc and Scientifica Inc) using a ×16 water immersion objective (Nikon, NA 0.8, LWD 3.0, plan). A mode-locked Ti:Sapphire laser (MaiTai Spectra-Physics) tuned to 920 nm was used for excitation. Either single plane or volumetric recording (6 planes with a Piezo) were obtained. Acquisition rates were for the recordings of *Tg(elavl3:GCaMP6s)*, and *Et (-0.6hsp70l:Gal4-VP16)s1020t;Tg(UAS:GCaMP6s)* fish: 40 Hz for a single plane of 1408 × 384 pixels or 3 Hz for a volume of 1536 × 900 pixels × 6 planes. For *Tg (GFAP:Gal4)nw7;Tg(UAS:GCaMP6s)* fish the acquisition rate was 24 Hz for a single plane of 1536 × 650 pixels. After a 3 min baseline period, a solution of 20 mM PTZ diluted in AFW was added through a perfusion system in order to induce epileptic activity. Total duration of these recordings were 60–70 min. For pilocarpine-induced seizures, a 60 mM solution of pilocarpine diluted in AFW was applied at the beginning of the recordings, which lasted for 120–240 min. Data analysis was done with a custom code written in MATLAB (Mathworks) as described in the following section.

**Data analysis**. Two-photon microscopy images were aligned using an adapted algorithm[30] that correct for occasional drift in the XY dimension, based on hierarchical model-based motion estimation. Individual neurons were automatically detected using a pattern recognition algorithm adapted from Ohki et al.[73], which identify neurons by using a correlation based approach comparing the GCaMP6s labeled neurons with torus or ring shaped neuronal templates[74]. Once all neurons were segmented they were individually tracked over the length of the whole recording to assure that the same neuron was properly captured. In case a neuron was lost during the tracking because of Z-drift, it was discarded from further analysis. Later, all automatically detected neurons are confirmed by visual inspection. Once the pixels belonging to each neuron were identified, the average of those pixels per image was calculated providing the complete time course of each individual neuron over time. Sparsely labelled glial cells were manually identified for Fig. 5 and large numbers of glial cells analyzed in Fig. 4 and Supplementary Fig. 7 were semi-automatically detected as described above for neurons. For each cell, the fractional change in fluorescence (ΔF/F) relative to the baseline was calculated[74]. All traces from neurons and glia were resampled to a final rate of 4 Hz (using decimate function in MATLAB). Neural and glial activity was studied in windows of 3 min: baseline (3 min prior to drug delivery), preictal period (3 min preceding the seizure) and seizure. In each interval a cell was considered active if its change in fluorescence was greater than three times the standard deviation from the baseline period. Generalized seizure onset was manually annotated as the time point with sudden increase of activity recruiting the majority of the cells in all brain regions. The activity of each cell was calculated using the area under the curve by trapezoidal numerical integration method (function trapz, MATLAB).

**Quantification and statistical analysis**. Statistical analysis was done using MATLAB. Wilcoxon ranksum test was used for non-paired analysis and Wilcoxon signed rank test for paired analysis. $P < 0.05$ was considered as statistically significant.

**Reporting summary**. Further information on research design is available in the Nature Research Reporting Summary linked to this article.

## Data availability
The datasets supporting the current study have not been deposited in a public repository, but are available from the corresponding author upon request.

## Code availability
The codes supporting the current study have not been deposited in a public repository, but are available from the corresponding author upon request.

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

## Acknowledgements

We thank M. Ahrens (HHMI, Janelia Farm, USA), C. Wyart (ICM, Paris, France), R. MacDonald and W. Harris (Cambridge University, UK), M. Orger (Champalimaud Centre for the Unknown, Lisbon, Portugal) and H. Baier (MPI, Martinsried, Germany) for transgenic lines, and T. Ohshima (Waseda university, Tokyo, Japan) for the *GFAP:Gal4* plasmid. We thank S. Eggen, M. Andresen, V. Nguyen and our fish facility support team for technical assistance. We also thank S. Neuhauss (University of Zurich), E. Brodtkorb (St. Olav's University Hospital and NTNU) and the Yaksi lab members for stimulating discussions. This work was funded by The Liaison Committee for Education, Research and Innovation in Central Norway ('Samarbeidsorganet') Grant (S.M-S., N.J-Y., E.Y.), Flanders Science Foundation (FWO) Grant (C.D.V., E.Y.), RCN FRIPRO Research Grant 239973 (E.Y.), ERC starting grant 335561 (R.P., E.Y.), and NBRP from Amed (K.K.). Work in the E.Y. lab is funded by the Kavli Institute for Systems Neuroscience at NTNU.

## Author contributions

Conceptualization, E.Y. and N.J-Y.; methodology, C.D.V., S.M-S., E.A., E.V.H., C.D., S.V., J.V., N.J-Y., E.Y.; data analysis, C.D.V., S.M-S., E.A., E.V.H., C.D., E.Y.; software, C.D.V., R.P., E.Y.; providing reagents and data, M.I.C., C.K., A.M., K.K.; investigation, all authors; writing, C.D.V., S.M-S., N.J-Y., E.Y.; review and editing, all authors; funding acquisition, S.M.-S., N.J-Y., E.Y.; supervision, N.J-Y., E.Y.

## Additional information

**Competing interests:** The authors declare no competing interests.

