## [Peer Review File · Nature Communications]

Reviewers' Comments:

Reviewer #1:

Remarks to the Author:

The authors demonstrate that calcium imaging can report similar temporal profiles to LFP recordings with single neuron resolution. This paper uses wide-field and 2-p calcium imaging in a PTZ model of seizure in zebrafish to demonstrate that neurons and glia become increasingly correlated during the pre-ictal and seizure periods. Finally the authors use channelrhodopsin in astrocytes to demonstrate that "activation" of glia can lead to changes in the membrane potential of nearby neurons. The paper is descriptive, but contains interesting information.

Main points

- 1.) The authors use a single model of acute seizure induction, PTZ. It is possible that the features of acute seizure that these authors observe such increases in certain brain region activity during the pre-ictal period and recruitment of glial activity are a unique feature of this single pharmacological model. To verify if these are indeed features of acute seizure or specific actions of PTZ the authors should test their main findings with additional models of seizure induction, such as pilocarpine, kainic acid, high potassium or NMDA. Is there a chronic model of seizures in fish?
- 2.) The authors point out that several brain regions show faster increases in pairwise correlation between neurons (Fig 3B & 3C). Are similar increases also present in those same brain regions for glia? Do these changes always proceed changes in neuronal activity?
- 3.) To demonstrate that glia are involved in the seizure generation the authors express ChR2 in astrocytes and demonstrate that membrane potential of nearby neurons is enhanced. ChR2 is widely used to depolarize neuronal membranes to generate action potentials. What does it mean that glia are "activated" by ChR2? What is the physiological significance of membrane depolarization in astrocytes as a seizure mimetic? Do such changes occur in seizure? In other words, are astrocytes depolarised during seizure? Experiments or discussion needed.
- 4.) In the discussion the authors argue that the synchronized glial activity during the pre-ictal period represents a protective glial function that reduces high levels of glutamate and breakdown of this function leads to seizure generation, however they provide no evidence of this in their paper. Furthermore it could be that the state change in astrocyte calcium activity is simply correlated with changes in neuronal activity, rather than causative.
- 5.) The authors should cite and discuss papers published by David Attwell and colleagues in Nature in the 1980-90s. These papers showed that glutamate uptake was affected by K⁺, Na⁺ and voltage. If these are all altered in seizure models, then this too may contribute. This needs to be discussed in line with glutamate gliotransmission.
- 6.) Add a time scale bar to Fig 5F and 5H.
- 7.) Most of the scatter graphs are too small to see properly. e.g Fig 4C-E, Fig 3B, Fig 2C,D,E,F. Please make them bigger. The asterixes are almost impossible to see.
8. 20mM PTZ seems very high. Please justify with experiments or citations to past use.

9. The discussion seems a bit long for such a small paper.

All of these points could be addressed with a revision.

Reviewer #2:

Remarks to the Author:

General comments

The findings reported by Verdugo and colleagues are interesting and timely, especially in light of the fact that the majority of studies over the past decade, with regard to the underlying mechanisms of seizure aetiology of genetic epilepsy animal models in particular, have been focused primarily on neuronal dysfunction, with much less known about the role glia play in the process of epileptogenesis. In addition, more recent investigation of several genetic epilepsy animal models as well as more recent observations obtained from differentiated patient-derived iPSCs have provided hints as to the important role that glia play in neuronal (dys)function.

Major comments

1. Although the results are certainly intriguing and novel, the study overall remains at the "descriptive" level and falls short of any attempts to address the mechanisms that could potentially regulate the reported glia-mediated state transitions. Thus, the results are still rather preliminary, as many questions remain unanswered. In addition, it would have been favourable to compare whether neuronal-glia interactions and transitions to the ictal phase observed in the PTZ seizure model also occur in a similar manner in at least one well characterised genetic epilepsy model. This comparison would address the question as to what happens when more localised, specific neuronal populations become dysfunctional (e.g. Gabaergic interneurons or Glutamatergic neurons) and how this would affect the neuronal-glia network.

2. Mechanisms regulating glial-neuronal interactions: This should be addressed, either through systematic testing of the effect of pharmacological inhibitors on transporters, gap junctions and ion channels (at least the "most likely suspects") in regulating coupling between neuronal and glial networks. What would happen for example if zebrafish were co-treated with Ifenprodil, an NMDAR antagonist that reduces glial-derived Glutamate? What about Connexin inhibitors? Alternatively, mutant zebrafish could replace pharmacological inhibitors, if these are available or if higher specificity is warranted.

3. Context and relevance of findings - and again, possible mechanisms: Any previous studies describing the role of astrocytes in the steps leading toward status epilepticus should be elaborated on in the discussion. Furthermore, in order to place the findings of this study within a larger context - especially when highlighting glia-neuron transitions, other examples of neuron-glia functional coupling should be mentioned (i.e. not just during seizures). For example, the role of connexins in functional coupling between neurons and astrocytes and how dysregulation plays a role in the pathogenesis of Parkinson's and Alzheimer's disease.

4. Mechanisms underlying neuronal synchronization: What other mechanisms regulate neuronal synchronization? Similar to question 3 above, in which other contexts (i.e. neurological disorders) does this happen? (Example, Alzheimer's disease). Context is important here as the authors argue for the described neuronal-glia interactions to be a more general phenomenon important for various transitions other than seizures.

5. Are gamma oscillations perturbed in the zebrafish PTZ model? Other frequencies?

Reviewer #3:

Remarks to the Author:

In this paper, the authors have used brain-wide imaging techniques to determine the activity of neurons and astrocytes across the brain during onset of seizures. The authors have found very interesting, state-dependent interactions between glia and neurons in the preictal and ictal period that may have implications for our understanding of seizure spread and generalization. The imaging results presented look beautiful, and I believe that applying them to the study of brain-wide synchronization and glioneuronal interaction is timely and interesting.

There are some obvious criticism of this paper that can be raised. Firstly, the authors pharmacologically block GABA receptors to induce seizures. This obviously means that – as responses to PTZ are concentration dependent (see also the Turrini et al. 2017 paper that is cited) – that the increasing PTZ concentration during washin will affect and shape the CNS responses that are measured during the different phases. Moreover, it is clear that pharmacologically induced seizures may be different from spontaneously arising seizures. I would like to explicitly say that I do not think that this invalidates the importance of the authors conclusions – the dramatic nature of the changes in glioneuronal activity and synchronization, in different, quite distinct stages of ictogenesis is still fascinating and important, even though this is a system that is under a general (and somewhat nonstationary) excitatory bias. I do think it is strange that the authors do not at all refer to this issue. They should, and they should explain exactly and concisely why they think (as I assume they do) their results have more general validity.

The second obvious criticism is that this is zebrafish, a brain far remote from the mammalian brain. I think this also is not a viable critique, as it is simply not possible to perform brain-wide imaging at cellular resolution in large brains. The authors could perhaps try to motivate better throughout the results the large advantages of brain-wide imaging, and perhaps also motivate better why they focus in on certain brain areas.

There are some additional issues I would like to raise:

One main finding is that neuronal correlations rise dramatically in a brain-wide fashion during ictal activity. The authors have used systematic pairwise Pearson correlations for this. This may be an issue, as there are a few pitfalls in analyzing time series correlations only with Pearson correlations. Firstly, slow shifts in baseline can generate correlations that are unrelated to the activity time scales that are actually under investigation. It would be important to ascertain that there are no such trends in some of the cells in the datasets (i.e. slow shifts) that generate correlations. If present, systematic models should be applied to remove trends. Secondly, there is a large increase in activity in the ictal states, that is then associated pretty rigidly with higher Pearson correlations. Are the authors sure that their correlations are not influenced by the i) amplitude of calcium signals or ii) within-time-series correlations that arise in the ictal conditions? To my knowledge, the Pearson correlation coefficient can be affected spuriously by both. Shuffled or surrogate data might provide a way to firm this up. Another issue is that the Pearson coefficient is really good at tracking linear correlations, but inefficient in detecting nonlinear ones. As one of the main statements the authors make is that glial activity is changing in the preictal period, subsequently giving rise to the strong synchronization of neurons in the ictal period, it would be important to show that the structure of the neuronal activity in the preictal state is indeed very different from the ictal one. Finally, there is a very high fraction of neuron pairs under some conditions that appear to have a Pearson coefficient of 1, meaning that they have perfect linear correlation. What are these cells? It is very hard to imagine that there would be real biological df/F signals that would be perfectly correlated. As a minor related point, I think it would be reasonable to show examples of single cell calcium traces during the different stages, to see what a high vs. low correlation actually looks like. The exact same argumentation, albeit perhaps at different

time scales would apply to the glial calcium measurements and their correlations.

Another main comment relates to additional questions one might ask of the cellular resolution dataset. Can spread of seizures be addressed more conclusively? It is one of the major topics of this paper, with substantial discussion of this issues in the introduction and discussion sections. I would really like to see if directional measures like Granger causality, or transfer entropy behave over time in this model. In particular, it would be interesting to see if directionality is observed only in specific stages of ictogenesis (vs. continuously). In addition, I would like to see some measures of delay between bulk signals of different regions (and then – if there is something interesting also for single-cell data). This is obviously limited by sampling, but the authors have a pretty decent sampling rate and it would in my opinion be worth a try.

Another issue is the more fine-grained temporal view of what happens before the seizure state. The authors show an example in Fig. 2B that raises some interesting points. Firstly, it seems as if there is a gradual incorporation of many cells in events that are very synchronous, but largely spare the telencephalon. These have some regularity, occurring every 20-30 seconds. I think it would be important to state i) if this is a general finding, ii) if so, with what average rates they occur and if there is a buildup before the seizure and iii) how many cells participate in which region. I would also like to see how much of the correlations (see above) is driven by these events, i.e. it would be important to compute the correlations measures taking out these definable events. The same would apply to the glia measurements, where there is a similar type of event (i.e. in Fig. 4B). Conversely, it would be useful to compute correlations using moving windows up to the ictal state between neurons, between glia but also for the neuron-glia experiments in Fig. 5. This would allow to discriminate episodic correlations that lead up to the strong correlations that denote a seizure.

The optogenetic experiments are important, and show that glia directly influence neurons that are in proximity to glial processes, but not those that are spatially distant. This is consistent with the views in the field, but there are some puzzling issues. Firstly, the voltage responses are extremely heterogeneous, at odds with the very precise and general synchronization the authors observe. Is there a methodical reason for this (i.e. differences in the optogenetic stimulation)? The authors should comment on this, even if the results overall are convincing. Secondly, the local nature of the glioneuronal interaction does not really explain the global synchronization via a glial mechanism that the authors claim. I think this should also be much more explicitly discussed in the discussion section. The authors have started to connect regional differences in PTZ susceptibility with GABAergic neuron density. I am not entirely sure whether this strengthens or weakens the story. It is certainly not really related to the glioneuronal synchronization mechanism. My feeling is that this relationship to neuronal numbers alone is very weak and not convincing, the efficacy of inhibition depends on many other additional factors. I recommend leaving this set of data, and the related discussion out of the paper. I also have a few remarks on the writing. The results section is very clear and beautifully written. In contrast, I feel that the introduction and discussion are less concise than they should be. As an example, the introduction contains an extended discussion of mechanisms of seizure initiation and spread. Many of these points (i.e. the idea of network hubs and choke points) are none the authors actually address, they appear again in the discussion. I think the authors should limit both the introduction and discussion to the salient points that they can support with data, and perhaps go through both sections carefully to shorten them and make them more concise.

Minor:

- The timepoints/intervals defining preictal should be given in the main text in the results, this is such a critical information given that this is an induced pharmacological model.
- The electrode position in fig. 1 should be specified in the legend
- The spectral analysis of fluorescent and LFP signals show that the fluorescence signals show much stronger decay of PSD towards higher frequencies, as expected for the slow off rate of the indicator. The authors should make clearer that all the panels E and F of Fig. 1 show is that both systems capture dynamics at the (in this case relevant) lower frequency ranges. It sounds a little bit as if the authors want this to stand as evidence of similarity between the signals.

- The similarity of LFP and bulk calcium signals is very evident in the example shown in Fig. 1C. Can this be quantified?
- The authors claim a decrease of high-power activity in Fig. 1E and F (line 141-142), but I can only see this in Fig. 1E for the LFP, it is the opposite for calcium, this should be corrected – any explanation for this?
- In Fig. 3D, there seem only to be positive correlations, while it is clear from Fig. 3E and F that there are negative ones. Is it true then that all negative correlations are for cell pairs $> 300 \mu\text{m}$ apart? And why is the dashed line for the shuffled locations indicating an average that is positive if all neurons were counted? Or is just a selection of pairs shown in Fig. 3D? This was not stated, I believe. The same applies for Fig. 4F.
- I would flip the color code in Fig. 3F to make the occurrence of the seizure red.
- In Fig. 5B, it would be important to label the occurrence of the seizure, is this at the right side of the plot? Or is this still preictal? It does not look the same intensity as the example seizure shown in Fig. 1B.

Reviewers' response

We thank all reviewers for their constructive feedbacks and for expressing their enthusiasm towards our work. We have now addressed all of the reviewer's comments in our revised manuscript. To this end, we added several new experiments and analyses, which we believe addressed all questions raised by the reviewers and increase the impact of our findings. In this letter, **the reviewers' comments are in red letters**, and our response to each comment is in black letters.

Kindest regards

Emre Yaksi and Nathalie Jurisch-Yaksi

Reviewer #1

The authors demonstrate that calcium imaging can report similar temporal profiles to LFP recordings with single neuron resolution. This paper uses wide-field and 2-p calcium imaging in a PTZ model of seizure in zebrafish to demonstrate that neurons and glia become increasingly correlated during the pre-ictal and seizure periods. Finally the authors use channelrhodopsin in astrocytes to demonstrate that "activation" of glia can lead to changes in the membrane potential of nearby neurons. The paper is descriptive, but contains interesting information.

We are happy to see that reviewer 1 finds our results interesting. We have now addressed the reviewer's comments in our revised manuscript. Specific answers to the reviewers' comments are indicated below.

Main points

1.) The authors use a single model of acute seizure induction, PTZ. It is possible that the features of acute seizure that these authors observe such increases in certain brain region activity during the pre—ictal period and recruitment of glial activity are a unique feature of this single pharmacological model. To verify if these are indeed features of acute seizure or specific actions of PTZ the authors should test their main findings with additional models of seizure induction, such as pilocarpine, kainic acid, high potassium or NMDA. Is there a chronic model of seizures in fish?

Our response R1#1: We thank the Reviewer 1 for these suggestions. In order to address this comment, we tried various alternative strategies to model generalized seizures. In our experience we did not observe generalized seizures with kainic acid. However, we successfully generate generalized seizures upon pilocarpine delivery similarly to the ones generated by PTZ, although with a longer time delay for the seizure onset. Our findings regarding the involvement of glial activity and synchrony preceding the generalized seizures are not only a feature of the PTZ model, but can be observed in another generalized seizure model. We have now included this new data from neural and glial activity and the associated analysis obtained with the pilocarpine induced seizure model in our revised manuscript, Supplementary Figures 4 and 7.

Furthermore, we attempted to investigate if our findings on neural and glial recruitment can be generalized to genetic/chronic epilepsy models, by using mutant zebrafish lines (e.g. *scn1lab* mutant which has been well characterized by Baraban *et al*, 2013). We have initiated the breeding of these mutant lines to our reporter zebrafish lines, but did not have sufficient time to obtain imaging data for the genetic mutants. Breeding of these lines to obtaining of homozygous mutant takes longer than 5-6 months, which was longer the 3 months allocated time for the revision by Nature Communications. We hope that our new experiments with pilocarpine induced seizures will convince the reviewers that our conclusions about the

features of glial and neural activity preceding and during the spread of generalized seizures are not only a feature of PTZ induced seizures, but can be generalizable to other seizure models. Investigations of these mechanisms in different genetic epilepsy models will be an exciting addition to our results in future studies focusing on genetic epilepsy models.

2.) The authors point out that several brain regions show faster increases in pairwise correlation between neurons (Fig 3B & 3C). Are similar increases also present in those same brain regions for glia? Do these changes always proceed changes in neuronal activity?

Our response R1#2: We have now performed additional analyses to specifically answer this comment. In order to investigate whether glial networks in different brain regions behave differently before seizures or not, we performed our analysis separately in the telencephalon and thalamus for the GFAP expressing glial cells. We observed that glia in the telencephalon are significantly more active and more correlated than the thalamic glial population during the pre-ictal period compared to baseline. We have now added this information in Supplementary Figure S5. Since the telencephalon is the last brain area to be recruited during the generalized seizure and the brain area showing the highest and more synchronized glial activity in the pre-ictal stage, these new results support the hypothesis that glial activity in the pre-ictal phase have a protective role and thus delay the spread of the seizure to the telencephalon. Despite these small and interesting differences between the glial populations of the telencephalon and the thalamus, we also observed that glial network activity preceding the generalized seizures is highly synchronized within and across the telencephalon and the thalamus independent of spatial distances between glial cells. This is likely due to strong gap junction coupling between glial cells, which we now demonstrated conclusively by the crossing of neurobiotin from individually filled glial cells to large number of glia in the brain, now presented in Figures 4J and 6B . Finally, we also added single-cell sequencing data from zebrafish glial cell, showing glia-specific connexin43 expression in zebrafish telencephalon, presented in Supplementary Figure S6.

3.) To demonstrate that glia are involved in the seizure generation the authors express ChR2 in astrocytes and demonstrate that membrane potential of nearby neurons is enhanced. ChR2 is widely used to depolarize neuronal membranes to generate action potentials. What does it mean that glia are “activated” by ChR2? What is the physiological significance of membrane depolarization in astrocytes as a seizure mimetic? Do such changes occur in seizure? In other words, are astrocytes depolarised during seizure? Experiments or discussion needed.

Our response R1#3: We thank the reviewer for raising this very interesting point. During the course of the review process, we performed several additional experiments that answer these important questions.

1. Our intracellular recordings of Chr2 expressing glial cells showed that optogenetic stimulation of glia directly and quickly depolarizes the glial membrane potential. We included these data in Figure 6A

2. Moreover, we simultaneously measured glial membrane potential and glial calcium levels during PTZ induced epileptic seizures. Our results showed that glial calcium activity and glial membrane potential are highly correlated during the preictal and ictal period, and thereby confirm that astrocytes are depolarized during epileptic seizures. We included these data in Figure 4C. We also showed that glial calcium signals and glial membrane potential are highly correlated Figure 4D.

Moreover, these experiments further revealed the strong gap junction coupling through neurobiotin crossing between large populations of glial cells (Figure 4J & 6B), which certainly bring a whole new value to our findings. We are grateful to the reviewer for this constructive advice.

4.) In the discussion the authors argue that the synchronized glial activity during the pre-ictal period represents a protective glial function that reduces high levels of glutamate and breakdown of this function leads to seizure generation, however they provide no evidence of this in their paper. Furthermore, it could be that the state change in astrocyte calcium activity is simply correlated with changes in neuronal activity, rather than causative.

Our response R1#4: We thank the reviewer for this constructive advice, which we now have addressed by multiple means.

1. To address the reviewers comment about glutamate, we measured glutamate levels during seizure generation, using a transgenic line expressing the glutamate sensor iGluSnfr in all glia (GFAP:iGluSnfr), together with simultaneous local field potential recordings to measure the epileptic seizures. Our new results revealed that the extracellular glutamate levels are relatively constant during preictal period, despite the elevated neural and glial activity, but showed a sharp rise followed by a depression only during generalized seizures detected by local field potential recordings. We included this new data in our revised manuscript in Figure 5F and further discussed it in our results.

2. To address the reviewers comment about the relationship between glial and neural activation during preictal and ictal periods, we reanalyzed our data with a specific focus on the temporal structure of glial and neural activity bursts during pre-ictal and ictal periods. These new analyses revealed that during the pre-ictal state, neural activity burst comes first and followed by the burst of glial activity. Excitingly we observed that glial activity bursts coincide with a strong reduction of neural activity. Hence bursts of neural and glial activity during pre-ictal period is completely anti-correlated, which highlights a relative reduction of neural activity exactly during the period of glial bursts. This picture however altered completely, once the first generalized seizure is initiated and following bursts of generalized seizures continue. We found that during the first generalized seizure a small neural activity preceded the glial activity, but the neural activity grew drastically as soon as the glial activity is initiated. During the period of continuous generalized seizures, glial and neural activity was initiated simultaneously. These new analyses showed a drastic change in the temporal relationship between glial and neural activity as animals move from a preictal to an ictal state. Hence, our new results showed that observed changes in glial activity are not simply a reflection of the changes in the neural activity, but glia-neuron interactions are drastically altered during the seizure generation process. We have added these results in Figure 5E and discussed further in the revised manuscript.

5.) The authors should cite and discuss papers published by David Attwell and colleagues in Nature in the 1980-90s. These papers showed that glutamate uptake was affected by K⁺, Na⁺ and voltage. If these are all altered in seizure models, then this too may contribute. This needs to be discussed in line with glutamate gliotransmission.

Our response R1#5: We thank the reviewer for this excellent reminder. We have now better integrated the work of Prof Attwell in our manuscript.

6.) Add a time scale bar to Fig 5F and 5H.

Our response R1#6: We included scale bars on these figures (which in the revised manuscript are figures 6D and 6F, respectively).

7.) Most of the scatter graphs are too small to see properly. e.g Fig 4C-E, Fig 3B, Fig 2C,D,E,F. Please make them bigger. The asterixes are almost impossible to see.

Our response R1#7: We increased the font size in all figures, as well as the line width of the dots on scatter plots to improve the readability of all figures.

8. 20mM PTZ seems very high. Please justify with experiments or citations to past use.

Our response R1#8: 20mM PTZ has been commonly used in the literature to induce seizure in zebrafish larvae, e.g. Baraban et al 2005 and Afrikanova et al 2013. We have now added 3 citations in the manuscript justifying this dose. We also included a Supplementary Figure S1, comparing the effect of various concentrations of PTZ from 5mM, 10mM and 20mM on the percentage of animals seizing within 1h of treatment, the numbers of seizures in 1h treatment and the onset of the first generalized seizure. These results motivated us to use 20mM since it reliably induces generalized seizure in more than 70% of animals, with a relatively short onset, which is key for our whole brain imaging experiments. We thank the reviewer for this suggestion, which we hope that will provide guidelines for future experiments of similar kind using PTZ in zebrafish.

9. The discussion seems a bit long for such a small paper.

Our response R1#9: We have edited our discussion and taken the reviewer's comment into consideration.

Reviewer #2 (Remarks to the Author):

The findings reported by Verdugo and colleagues are interesting and timely, especially in light of the fact that the majority of studies over the past decade, with regard to the underlying mechanisms of seizure aetiology of genetic epilepsy animal models in particular, have been focused primarily on neuronal dysfunction, with much less known about the role glia play in the process of epileptogenesis. In addition, more recent investigation of several genetic epilepsy animal models as well as more recent observations obtained from differentiated patient-derived iPSCs have provided hints as to the important role that glia play in neuronal (dys)function.

We are delighted to see that reviewer 2 finds our results timely and interesting. We have now addressed the specific comments of the reviewer in our revised manuscript as described below.

Major comments

1. Although the results are certainly intriguing and novel, the study overall remains at the "descriptive" level and falls short of any attempts to address the mechanisms that could potentially regulate the reported glia-mediated state transitions. Thus, the results are still rather preliminary, as many questions remain unanswered. In addition, it would have been favourable to compare whether neuronal-glia interactions and transitions to the ictal phase observed in the PTZ seizure model also occur in a similar manner in at least one well characterised genetic epilepsy model. This comparison would address the question as to what happens when more localised, specific neuronal populations become dysfunctional (e.g. Gabaergic interneurons or Glutamatergic neurons) and how this would affect the neuronal-glia network.

Our response R2#1: We thank the reviewer #2 for this excellent suggestion, which is also in line with the first comment of the reviewer 1. To address this comment, we used pilocarpine (another well established compound for seizure generation) to validate that our findings regarding the involvement of glial activity and synchrony preceding the generalized seizures are not only a feature of the PTZ model. We have now included this new data and analysis

from neural and glial activity and the associated analysis obtained with the pilocarpine induced seizure model in our revised manuscript, Supplementary Figures 4 and 7. Furthermore, we attempted to investigate if our findings on neural and glial recruitment can be generalized to genetic/chronic epilepsy models, by using mutant zebrafish lines (e.g. *scn1lab* mutant which has been well characterized). We have initiated the breeding of these mutant lines to our reporter zebrafish lines, but did not have sufficient time to obtain imaging data for the genetic mutants. Breeding of these lines to obtaining of homozygous mutant takes longer than 5-6 months, which was longer than allocated time (3 months) for the revision by Nature Communications. We hope that the reviewers will be convinced that our conclusions about the features of glial and neural activity preceding and during the spread of generalized seizures are not only a feature of PTZ induced seizures, but can be generalizable to other seizure models. Investigations of these mechanisms in different genetic epilepsy models will be an exciting addition to our results in future studies focusing on genetic epilepsy models.

2. Mechanisms regulating glial-neuronal interactions: This should be addressed, either through systematic testing of the effect of pharmacological inhibitors on transporters, gap junctions and ion channels (at least the “most likely suspects”) in regulating coupling between neuronal and glial networks. What would happen for example if zebrafish were co-treated with Ifenprodil, an NMDAR antagonist that reduces glial-derived Glutamate? What about Connexin inhibitors? Alternatively, mutant zebrafish could replace pharmacological inhibitors, if these are available or if higher specificity is warranted.

Our response R2#2: We agree with Reviewer 2 with respect to the value of further investigating glia-neuron interactions, and thank for the suggestions. We performed several new experiments and analysis, which certainly increase the impact and value of our findings.

1. To characterize glia-neuron interactions further, we combined the channelrhodopsin-mediated activation of glia with electrophysiological recordings of neurons and pharmacology. First, we showed that optogenetic activation directly depolarizes glial membrane potential (Figure 6A). After performing hundreds of new electrophysiological recordings, we identified that not only Chr2 expressing glia can activate nearby neurons, but also showed that neuronal excitation upon optogenetic activation of glia follows at least 2 different types, a strong connection with fast kinetics, and a medium/weak connection with slow activation kinetics (Figure 6H).
2. Inspired with this diversity of glial activation of neurons we first tested ionotropic glutamate receptor blockers NBQX/AP5 and found out that the neural activity with medium/weak connection strength and slow kinetics were sensitive to NBQX/AP5, highlighting a glutamate dependent part of glia-neuron excitation (Figure 6I,J).
3. However we also found out that some glia to neuron excitation with high amplitude and fast kinetics were still significant even after the addition of NBQX/AP5 (Figure 6J). This bring about the possibility that some of these fast connections between glia and neurons might be through the action of gap junctions. Hence, we performed additional experiments by using the gap junction blocker carbenoxolone. In fact, blocking gap junctions with carbenoxolone reduced the strength of glia-neuron coupling (Figure 6 K,L). We also further discuss these new results.
4. Moreover, inspired by suggestion of the reviewer #2, we also tested the effect of Ifenprodil, with the hope to see a significant effect in the characteristics of epileptic seizure generation. Unfortunately, we did not see any significant effect by using Ifenprodil in any of the seizure generation parameters that we investigated. Average time to seizure onset: ifenprodil 17,8857 +/- 6,9061 min, control: 16,6750 +/- 12,1322 min. Average duration of the first seizure: ifenprodil 0,4429 +/- 0,1512 min, control 0,3500 +/- 0,3109 min (mean +/- SD). Since these experiments did not show any prominent effect with ifenprodil application we

decided not to include them in the manuscript. However, we are happy to include this new data if the reviewer #2 thinks that it is important.

All these additional experiments certainly bring further value to our findings and provide a better mechanistic explanation for the glia-neuron communications, which involve both glutamate but also gap junctions. In fact, it would be very interesting to investigate the role of these connections further by using genetic strategies, which require creation of new transgenic lines. Since our new results showed strong neurobiotin (gap junction) coupling between glia (Figure 4J + 6B) as well as the specific expressions of Connexin 43 in zebrafish glial cells (Supplementary Figure S6), Connexin 43 would be an excellent target to specifically target the function of gap junctions, which we hope to address further in future studies

3. Context and relevance of findings - and again, possible mechanisms: Any previous studies describing the role of astrocytes in the steps leading toward status epilepticus should be elaborated on in the discussion. Furthermore, in order to place the findings of this study within a larger context - especially when highlighting glia-neuron transitions, other examples of neuron-glia functional coupling should be mentioned (i.e. not just during seizures). For example, the role of connexins in functional coupling between neurons and astrocytes and how dysregulation plays a role in the pathogenesis of Parkinson's and Alzheimer's disease.

Our response R2#3: We thank the Reviewer2 for these suggestions and we now extend our discussion on this matter. Indeed, investigations of glia neuron interactions as well as gap junctions are getting increasingly popular in many different neurological phenomena and diseases. We discussed these different phenomena more rigorously in the revised manuscript. Since the Reviewer 3 specifically asked us to shorten the introduction and the discussion section (please see "Our response R3#13" below), we added the discussion suggested by Reviewer 2, while trying to be concise.

4. Mechanisms underlying neuronal synchronization: What other mechanisms regulate neuronal synchronization? Similar to question 3 above, in which other contexts (i.e. neurological disorders) does this happen? (Example, Alzheimer's disease). Context is important here as the authors argue for the described neuronal-glia interactions to be a more general phenomenon important for various transitions other than seizures.

Our response R2#4: We thank the Reviewer2 for this remark, and indeed we fully agree about the general importance of neuronal-glia interactions and neural synchrony in brain function during both health and disease. We now extended our discussion on this topic in our revised manuscript and try to keep a balance between being more concise but also covering broadly these suggestions.

5. Are gamma oscillations perturbed in the zebrafish PTZ model? Other frequencies?

Our response R2#5: We investigated the power spectrum of our local field potential recordings presented in Fig1D more carefully. We did neither observe prominent gamma oscillations in larval zebrafish (5 days old), nor changes in the gamma frequency range between preictal and ictal period. Instead we observed most of the changes in the 0.1-1Hz frequency range during the pre-ictal and ictal period, suggesting that gamma oscillations are not prominent in our larval zebrafish PTZ model. We have discussed these results better in the revised manuscript.

Reviewer #3 (Remarks to the Author):

In this paper, the authors have used brain-wide imaging techniques to determine the activity of neurons and astrocytes across the brain during onset of seizures. The authors have found very interesting, state-dependent interactions between glia and neurons in the preictal and

ictal period that may have implications for our understanding of seizure spread and generalization. The imaging results presented look beautiful, and I believe that applying them to the study of brain-wide synchronization and glioneuronal interaction is timely and interesting.

We thank the reviewer for her/his compliments on our results. In this new revised manuscript, we believe that we answered all important points raised by the reviewer #3

There are some obvious criticism of this paper that can be raised. Firstly, the authors pharmacologically block GABA receptors to induce seizures. This obviously means that – as responses to PTZ are concentration dependent (see also the Turrini et al. 2017 paper that is cited) – that the increasing PTZ concentration during washin will affect and shape the CNS responses that are measured during the different phases. Moreover, it is clear that pharmacologically induced seizures may be different from spontaneously arising seizures. I would like to explicitly say that I do not think that this invalidates the importance of the authors conclusions – the dramatic nature of the changes in glioneuronal activity and synchronization, in different, quite distinct stages of ictogenesis is still fascinating and important, even though this is a system that is under a general (and somewhat nonstationary) excitatory bias. I do think it is strange that the authors do not at all refer to this issue. They should, and they should explain exactly and concisely why they think (as I assume they do) their results have more general validity.

Our response R3#1: As stated by the Reviewer 3 and all other reviewers above, it is indeed very valuable to investigate if the phenomena that we observe can be extended to other zebrafish models of epilepsy ideally with generalized seizures. As described in more detail in the responses to other reviewers (please see **Our response R1#1**), we performed additional experiments in zebrafish treated with the proconvulsant pilocarpine that acts on muscarinic acetylcholine receptor, unlike PTZ that acts on GABAergic receptors. In these pilocarpine induced generalized seizures we have observed similar changes in neuronal and glial activity and synchrony (Supplementary Figures 4 and 7). We hope that these additional experiments increase the general validity of our observations for epilepsy. Investigations of these mechanisms in different genetic epilepsy models will be an exciting addition to our results in future studies focusing on genetic epilepsy models.

The second obvious criticism is that this is zebrafish, a brain far remote from the mammalian brain. I think this also is not a viable critique, as it is simply not possible to perform brain-wide imaging at cellular resolution in large brains. The authors could perhaps try to motivate better throughout the results the large advantages of brain-wide imaging, and perhaps also motivate better why they focus in on certain brain areas.

Our response R3#2: We thank Reviewer3 for reminding us this very valid point. We better justified the use of zebrafish for this study. We also better highlighted the conservation of several vertebrate brain structures also in zebrafish.

There are some additional issues I would like to raise: One main finding is that neuronal correlations rise dramatically in a brain-wide fashion during ictal activity. The authors have used systematic pairwise Pearson correlations for this. This may be an issue, as there are a few pitfalls in analyzing time series correlations only with Pearson correlations. Firstly, slow shifts in baseline can generate correlations that are unrelated to the activity time scales that are actually under investigation. It would be important to ascertain that there are no such trends in some of the cells in the datasets (i.e. slow shifts) that generate correlations. If present, systematic models should be applied to remove trends. Secondly, there is a large increase in activity in the ictal states, that is then associated pretty rigidly with higher Pearson

correlations. Are the authors sure that their correlations are not influenced by the i) amplitude of calcium signals or ii) within-time-series correlations that arise in the ictal conditions? To my knowledge, the Pearson correlation coefficient can be affected spuriously by both. **Shuffled** or surrogate data might provide a way to firm this up.

Our response R3#3: We thank to Reviewer3 for these suggestions. In all our imaging data we used a systematic running baseline calculation and subtraction, which corrects all slow shifts and trends in baseline neural activity. We now clarified this better in the text. We are intrigued by the suggestion of the Reviewer3 with respect to our use of Pearson's Correlations as a metric for similarity. In principle Pearson's Correlations are calculated after subtracting mean and dividing by the standard deviations (zscores), which should reduce the influence of amplitude, and highlighting the shape of neural activity (for example unlike Euclidean distances). Hence to the best of our knowledge, calculating Pearson's correlations is a standard approach to investigate similarities/synchrony of neural activity (time-series) in most studies investigating multi-neuronal activity/synchrony (some reference studies here: DOIs : [10.1038/nmeth.2434](https://doi.org/10.1038/nmeth.2434), [10.1016/j.cub.2011.12.002](https://doi.org/10.1016/j.cub.2011.12.002), [10.1073/pnas.1521299113](https://doi.org/10.1073/pnas.1521299113), [10.1523/JNEUROSCI.4007-14.2015](https://doi.org/10.1523/JNEUROSCI.4007-14.2015)).

Furthermore, indeed the sudden drastic increase of neural activity in thousands of recorded neurons is likely the main source of increased synchrony that we observe during the ictal stage, as all neurons simultaneously increase their activity with similar kinetics during the ictal period. However, we agree with the Reviewer 3 that it is more convincing to shuffle the time series data from individual neurons during the ictal stage and show that the Pearson's correlations are reduced. Taking the advice of the reviewer #3, we shuffled the time series of individual neurons and re-run our correlation-based analysis. We observed that shuffling time series of individual neurons in all stages (baseline, preictal, ictal) completely reduced the correlations between neuronal activity (Supplementary Figure S3E,F,G,H), which demonstrated that the Pearson's correlations we observed between neurons were not simply due to the amplitude of calcium signals. We thank the reviewer #3 for suggesting this important control and we hope that this analysis further clarified the validity of our arguments.

Finally, we also analyze our data in different ways (other than correlations) to highlight the strong change between the relationship of neuronal and glial activity during the pre-ictal and ictal stages, please see **Our response R3#4** just below for further explanation.

Another issue is that the Pearson coefficient is really good at tracking linear correlations, but inefficient in detecting nonlinear ones. As one of the main statements the authors make is that glial activity is changing in the preictal period, subsequently giving rise to the strong synchronization of neurons in the ictal period, it would be important to show that the structure of the neuronal activity in the preictal state is indeed very different from the ictal one.

Our response R3#4: In fact we now performed further analysis of individual glial and neuronal bursts during the pre-ictal and ictal stage (Figure 5E). These new analyses showed that the relationship between glial and neuronal activity drastically differs not only in the correlations but also in the temporal relationships, during neuronal and glial bursts. In summary, we observed that during the preictal period, the neuronal activity bursts preceded glial bursts. Interestingly glial activity bursts coincide with a strong reduction of neural activity during the preictal stage, which potentially highlights the protective action of glial activity during this stage. This picture however altered completely, once the first generalized seizure is initiated and the following bursts of generalized seizures continue. We found that during the first generalized seizure a small neural activity preceded the glial activity, but the neural activity grew drastically as soon as the glial activity is initiated. During the period of

continuous generalized seizures, glial and neural activity was initiated simultaneously. These new analyses showed a drastic change in the temporal relationship between glial and neural activity as animals move from a preictal to an ictal state. We hope that these new analyses better reveal the structure of neural and glial activity, during the transition from pre-ictal to ictal stage.

Finally, there is a very high fraction of neuron pairs under some conditions that appear to have a Pearson coefficient of 1, meaning that they have perfect linear correlation. What are these cells? It is very hard to imagine that there would be real biological df/F signals that would be perfectly correlated.

Our response R3#5: This is due to large bin size of our histograms representing these correlations. We looked at our data in more depth and we can confirm that we don't observe Pearson's correlation coefficients of 1. The maximum value we obtained for correlation was 0.9989.

As a minor related point, I think it would be reasonable to show examples of single cell calcium traces during the different stages, to see what a high vs. low correlation actually looks like. The exact same argumentation, albeit perhaps at different time scales would apply to the glial calcium measurements and their correlations.

Our response R3#6: We now provide calcium traces examples from individual neurons and glial cells in our revised manuscript. Please find these example traces in Supplementary Figure S3A,B,C,D. We appreciate this suggestion, as displays some of the concepts of this manuscript, and the quality of our imaging data more clearly.

If the reviewer #3 wishes to inspect further, please also find below the precise correlation coefficient values of individual neurons that are represented in the correlation matrices in this Supplementary Figure S3D.

Baseline	Cell #1	Cell #2	Cell #3	Cell #4	Pre-ictal	Cell #1	Cell #2	Cell #3	Cell #4	Ictal	Cell #1	Cell #2	Cell #3	Cell #4
Cell #1	1	0.13	0.29	-0.15	Cell #1	1	0.52	0.81	-0.04	Cell #1	1	0.93	0.91	0.51
Cell #2	0.13	1	0.16	0.02	Cell #2	0.52	1	0.68	0.12	Cell #2	0.93	1	0.84	0.26
Cell #3	0.29	0.16	1	-0.1	Cell #3	0.81	0.68	1	0.17	Cell #3	0.91	0.84	1	0.41
Cell #4	-0.15	0.02	-0.10	1	Cell #4	-0.04	0.12	0.17	1	Cell #4	0.51	0.26	0.41	1

Another main comment relates to additional questions one might ask of the cellular resolution dataset. Can spread of seizures be addressed more conclusively? It is one of the major topics of this paper, with substantial discussion of this issues in the introduction and discussion sections. I would really like to see if directional measures like Granger causality, or transfer entropy behave over time in this model. In particular, it would be interesting to see if directionality is observed only in specific stages of ictogenesis (vs. continuously). In addition, I would like to see some measures of delay between bulk signals of different regions (and then – if there is something interesting also for single-cell data). This is obviously limited by sampling, but the authors have a pretty decent sampling rate and it would in my opinion be worth a try.

Our response R3#7: We thank the reviewer for these suggestions to further investigate the spread of the generalized seizures. We agree that investigating directionality and time delays would add up to the richness of our findings. Our previous attempts using Granger causality in our calcium imaging data sets suggest that the temporal resolution is not sufficiently good (since the peak cross correlations were at zero delay). To investigate time delays across different brain regions during seizure generalization, we investigated the bulk signal average

of hundreds of individual neurons in identified brain areas in all our recorded animals Supplementary Figure S2A. Our new analysis comparing the onset of calcium signals during generalized seizure did not reveal a significant time delay between most brain areas, except the telencephalon (Supplementary Figure S2B). In fact, we observed that the telencephalon was the only brain area that displayed a significant delay during the generalized seizure onset, consistently following the activity of other brain regions.

These new results also in line with our other measurements of neural activity and neural synchrony in the telencephalon during the pre-ictal activity in comparison to other brain regions (Figure 2 D,F and Figure 3B). Interestingly, our new analysis, separating glial activity in telencephalon and the thalamus showed that telencephalic glial populations shows significant changes in their activity and synchrony, and less for thalamic glial populations (Supplementary Figure S5). Combined with our results on the individual neural and glial activity bursts during pre-ictal period (Figure 5E), all these results suggest that one reason that the neuronal activity of telencephalon is less effected during the pre-ictal period, and joins the generalized seizures with a delay, might be due to the protective effect of pre-ictal glial activity in this brain region. We now better discussed these new results in our manuscript.

Another issue is the more fine-grained temporal view of what happens before the seizure state. The authors show an example in Fig. 2B that raises some interesting points. Firstly, it seems as if there is a gradual incorporation of many cells in events that are very synchronous, but largely spare the telencephalon. These have some regularity, occurring every 20-30 seconds. I think it would be important to state i) if this is a general finding, ii) if so, with what average rates they occur and if there is a buildup before the seizure and iii) how many cells participate in which region.

Our response R3#8: It is true that the telencephalon is always an outsider when compared to other brain regions, both for the amount of neural activity and for the correlation of neural time series, which we reported in our manuscript, also now added more results and discussions (as explained in **Our response R3#7**). Intrigued by the suggestion of the reviewer 3, we inspected the regularity and the periodicity of the neural bursts during the preictal state. In average, across n=8 fish, these pre-ictal burst occur every 97.48 seconds; although with a standard deviation of 137.47 seconds. We can confirm that this was not a general feature, and such inter-burst intervals are highly variable across animals. Therefore, we chose not to further inspect the periodicity and features of the bursts. Instead we investigated these individual bursts of activity in those experiments that we could image both neurons and glia simultaneously. The results of such analysis on identified bursts of activity is explained more in depth in **Our response R3#4**.

I would also like to see how much of the correlations (see above) is driven by these events, i.e. it would be important to compute the correlations measures taking out these definable events. The same would apply to the glia measurements, where there is a similar type of event (i.e. in Fig. 4B). Conversely, it would be useful to compute correlations using moving windows up to the ictal state between neurons, between glia but also for the neuron-glia experiments in Fig. 5. This would allow to discriminate episodic correlations that lead up to the strong correlations that denote a seizure.

Our response R3#9: Calcium signals are not good in detecting very small changes of neural activity, unless few neurons are imaged at very high spatial and temporal resolution. Hence, we believe that most of the correlations we observe are due to these relatively large events highlighted by the Reviewer3. As shown in Figure 5E, we identified these individual episodic bursts and inspect the temporal relations of neural and glial activity, in addition to the episodic correlations of neural and glial activity. We explained these features in more detail in

Our response R3#4. Our new findings during the analysis of these episodic burst certainly added up to our main conclusions of the manuscript. We thank the reviewer for this suggestion.

The optogenetic experiments are important, and show that glia directly influence neurons that are in proximity to glial processes, but not those that are spatially distant. This is consistent with the views in the field, but there are some puzzling issues. Firstly, the voltage responses are extremely heterogeneous, at odds with the very precise and general synchronization the authors observe. Is there a methodical reason for this (i.e. differences in the optogenetic stimulation)? The authors should comment on this, even if the results overall are convincing.

Our response R3#10: Indeed, the heterogeneity of the neuronal voltage response amplitude up on glial optogenetic stimulation is mainly methodological. Our zebrafish line, has only sparse and rather patchy expression of channelrhodopsin2 (Chr2) in glial cells. This means that we have no control on how many glial cells will express Chr2, and how much of the processes of randomly recorded neurons will pass through glial arborizations. This leads to quite some heterogeneity in the neuronal voltage response amplitude. We explained this better in the revised manuscript.

Additionally, during the course of the revision process, we recorded hundreds of new neurons while activating glia and found out over 43 neurons that showed significant excitation upon optogenetic glial activation. Analyzing these results further revealed another interesting diversity for the onset delay of neural excitation and its relation to the response amplitude (Figure 5E). We found out that neurons with strong excitation amplitude usually showed very short delay, whereas neurons with small but significant excitation showed rather long excitation onset delays, upon glial activation. We interpret these results as, neurons with strong excitation and short delays are very close to glial patches expressing Chr2, whereas neurons with weak and delayed excitations are further away from glial patches and might even receive indirect excitation, through gap junction coupling of glial cells (as evident in Figure 4J and 6B). We added these results and discussion further in our main text.

Secondly, the local nature of the glioneuronal interaction does not really explain the global synchronization via a glial mechanism that the authors claim. I think this should also be much more explicitly discussed in the discussion section.

Our response R3#11: We thank for this suggestion, we now have a much better understanding of these phenomenon after our new experiments during the revision. Our new results showed that glial cells in zebrafish express Connexin 43 (Supplementary Figure S6). Moreover, we observed very strong neurobiotin coupling between glial cells (examples in Figure 4J and 6B), which highlights a highly coupled glial network across very large distances in the brain, similar to astrocytes in mammals. Together with our observation of highly synchronized glial calcium signals (Figure 4H), all these new results are in line with the potential role of this highly coupled glial network to spread excitation in the brain, during a generalized seizure. Hence despite the local glia-neuron interactions, global glia-glia interactions can contribute to the global spread of excitation. We now better explained these results and discuss them in the main text.

The authors have started to connect regional differences in PTZ susceptibility with GABAergic neuron density. I am not entirely sure whether this strengthens or weakens the story. It is certainly not really related to the glioneuronal synchronization mechanism. My feeling is that this relationship to neuronal numbers alone is very weak and not convincing,

the efficacy of inhibition depends on many other additional factors. I recommend leaving this set of data, and the related discussion out of the paper.

Our response R3#12: We agree with the reviewer, and decided to leave these claims out of the manuscript. Yet, we still decided to keep the figure displaying distribution of GABAergic and glutamatergic neurons to highlight the diversity of neuronal populations across the brain areas of zebrafish brain, which might underlie why different brain regions might behave differently in response to both PTZ and pilocarpine induced seizures. If the reviewer 3, still think that it is important to complete leave the Figure 2G out, we will be happy to do so.

I also have a few remarks on the writing. The results section is very clear and beautifully written. In contrast, I feel that the introduction and discussion are less concise than they should be. As an example, the introduction contains an extended discussion of mechanisms of seizure initiation and spread. Many of these points (i.e. the idea of network hubs and choke points) are none the authors actually address, they appear again in the discussion. I think the authors should limit both the introduction and discussion to the salient points that they can support with data, and perhaps go through both sections carefully to shorten them and make them more concise.

Our response R3#13: We wished to cite the important contributions of people on the spreading of epileptic seizures. Therefore, we included the idea of the hubs/choke points. However, we agree with the reviewer 3 that perhaps this aspect was highlighted too strongly. We now limit these arguments on some of these points in the introduction and in the discussion.

Minor:

- The timepoints/intervals defining preictal should be given in the main text in the results, this is such a critical information given that this is an induced pharmacological model.

Our response R3#14: We included the time intervals used in all analyses on the specific figures to help the reader.

- The electrode position in fig. 1 should be specified in the legend.

Our response R3#15: We included this information in the figure legend.

- The spectral analysis of fluorescent and LFP signals show that the fluorescence signals show much stronger decay of PSD towards higher frequencies, as expected for the slow off rate of the indicator. The authors should make clearer that all the panels E and F of Fig. 1 show is that both systems capture dynamics at the (in this case relevant) lower frequency ranges. It sounds a little bit as if the authors want this to stand as evidence of similarity between the signals.

Our response R3#16: We thank the reviewer for this reminder. Indeed, calcium imaging is good in capturing signals only in lower frequency ranges and not as effective as LFP recordings for detecting high frequency activity. We now clarified this better in the main text.

- The similarity of LFP and bulk calcium signals is very evident in the example shown in Fig. 1C. Can this be quantified?

Our response R3#17: We agree with the reviewer about the quantification of this similarity. In fact, we quantified the similarity of LFP and calcium signals in Fig1G by using the measure of coherence, which quantify that similarity at each frequency range. If the reviewer find the measure of coherence not sufficiently quantitative and has other suggestions for quantifying this phenomenon, we will be happy to add these quantifications too.

• The authors claim a decrease of high-power activity in Fig. 1E and F (line 141-142), but I can only see this in Fig. 1E for the LFP, it is the opposite for calcium, this should be corrected – any explanation for this?

Our response R3#18: We thank the reviewer for pointing this out. Indeed, we saw a slight decrease in high power activity only in LFP recordings not in calcium imaging, and it is only shown in Fig1D and E and not Fig1F. We apologize for this mistake and corrected this in the text

• In Fig. 3D, there seem only to be positive correlations, while it is clear from Fig. 3E and F that there are negative ones. Is it true then that all negative correlations are for cell pairs > 300 μm apart? And why is the dashed line for the shuffled locations indicating an average that is positive if all neurons were counted? Or is just a selection of pairs shown in Fig. 3D? This was not stated, I believe. The same applies for Fig. 4F.

Our response R3#19: We thank the reviewer for this point. Indeed, while we observe both positive and negative correlations, as evident in the histograms on Fig3E, we have slightly more positive correlations than negative correlation. This is due to the nature of calcium imaging where positive correlations are slightly easier to pickup than negative correlations (since with calcium signals excitation is better represented than inhibition). Hence, the shuffled distributions in Fig3D, which represent the average of all those correlations, are slightly above zero, during baseline and preictal periods.

Moreover, Fig3D also represents the average of all correlations, and the last bin on the right includes all correlations that are 300 microns and above. We clarified this by adding ≤ 300 in the last bin of the Figure 3D, and Figure 4F.

• I would flip the color code in Fig. 3F to make the occurrence of the seizure red.

Our response R3#20: We changed the colorbar so that the seizure appears red in order to be consistent with all other figures.

• In Fig. 5B, it would be important to label the occurrence of the seizure, is this at the right side of the plot? Or is this still preictal? It does not look the same intensity as the example seizure shown in Fig. 1B.

Our response R3#21: We included the time intervals used in all analyses on the specific figures to help the readers. We also indicated the onset of the seizure with an arrow on each figure

Reviewers' Comments:

Reviewer #1:

Remarks to the Author:

The authors have addressed my previous concerns with new experiments and/or with thoughtful comments. I have no additional changes to suggest. However, the grammar is still a little odd at times and the editors and authors should spend some time correcting this before the paper is published.

Reviewer #3:

Remarks to the Author:

I enjoyed reading the revised Version of this paper, and would like to commend the authors on the very thorough and rigorous Approach taken in Response to the Review comments. I realize that this was substantial additional experimentation and analysis. I have no further comments.

REVIEWERS' COMMENTS:

Reviewer #1 (Remarks to the Author):

The authors have addressed my previous concerns with new experiments and/or with thoughtful comments. I have no additional changes to suggest. However, the grammar is still a little odd at times and the editors and authors should spend some time correcting this before the paper is published.

Reviewer #3 (Remarks to the Author):

I enjoyed reading the revised Version of this paper, and would like to commend the authors on the very thorough and rigorous Approach taken in Response to the Review comments. I realize that this was substantial additional experimentation and analysis. I have no further comments.

We thank the reviewers for their invaluable feedback, and we are delighted to read that they find our revisions satisfactory. In accordance with the comment of Reviewer 1, we have revised the manuscript with the aim to improve the grammar.